# Social status and the relationship between income rank and well-being in 109 nations

Edika Quispe-Torreblanca [1]✉, Jan-Emmanuel De Neve [2] & Gordon D. A. Brown [3]

Well-being is linked to income. However, lower well-being among lower-income individuals may reflect either economic relative deprivation or the lower social status associated with a lower income rank. Here, using Gallup World Poll data from 109 countries and over 90,000 individuals, we test a general model that includes both relative income deprivation and income rank as special cases. In 80% of countries, subjective well-being is more strongly associated with within-nation rank of income than with absolute income or relative income deprivation. Income rank coefficients are over three times larger in the most materialistic countries, but smaller in countries with higher social capital: In countries with the highest civic engagement, the association between income rank and well-being is about 80% smaller. Results replicated in multiple survey years and are consistent with a link between income-related social status and subjective well-being that is stronger when social capital is low.

Are people with higher incomes happier? And, if they are, why are they happier? Many studies have examined the role of absolute income (e.g., refs. 1–3). However, other research has converged on the idea that what matters is relative income and that people's subjective well-being (SWB) is linked to how their incomes compare to the incomes of other people. But relative income can itself be conceptualized in different ways. One possibility is that people compare their income to the average of a set of comparison incomes (e.g., ref. 4); a second is that income-related relative deprivation matters[5], and a third is that people care about the relative ranked position of their income within a social comparison set[6]. These possibilities carry different psychological interpretations, implicate different mechanisms of social comparison, and differ in their implications for public policies that target SWB.

Several studies support a rank-based approach to the income-SWB link. It is widely suggested that people's SWB is associated with the relative ranked position of their income (e.g., refs. 6–9). Income rank is typically operationalized as the ordinal position of an individual's income within the relevant national income distribution, scaled between 0 and 1. Links between income rank and SWB have been found in single-nation studies in at least 8 different countries and in one study of 24 countries (ref. 10, see Supplementary Note 1).

The income rank hypothesis of SWB explains why country-level inequality has little or no association with aggregate SWB, at least in high-income countries (e.g., ref. 11) and why a given increment in income leads to a greater improvement in SWB in more equal countries[12]. Moreover, many approaches to subjective socio-economic status and its relation to health assign a central role to an individual's rank within society[13]. If income rank is a status indicator, the income rank hypothesis is consistent with evidence that people engage in social comparison and care about their social status (e.g., ref. 14). The income-rank account also fits well with rank-based cognitive process models of the psychology of judgment[15].

However, most research has failed to distinguish between relative income deprivation and social status accounts of income's links to SWB. Social status accounts of income and well-being assume that what matters is the relative ranked position of an individual's income within a social comparison set and hence that the number of higher (and lower) earners will be all that matters, independently of how much more the higher earners receive. For example, if a person earns $20,000, the existence of other people earning $30,000 and $40,000 will have the same effect on that person's SWB as if the other people were earning $40,000 and $50,000. Measures of

[1]Leeds University Business School, University of Leeds, Leeds, UK. [2]Saïd Business School, University of Oxford, Oxford, UK. [3]Department of Psychology, University of Warwick, Coventry, UK. ✉e-mail: E.Quispe-Torreblanca@leeds.ac.uk

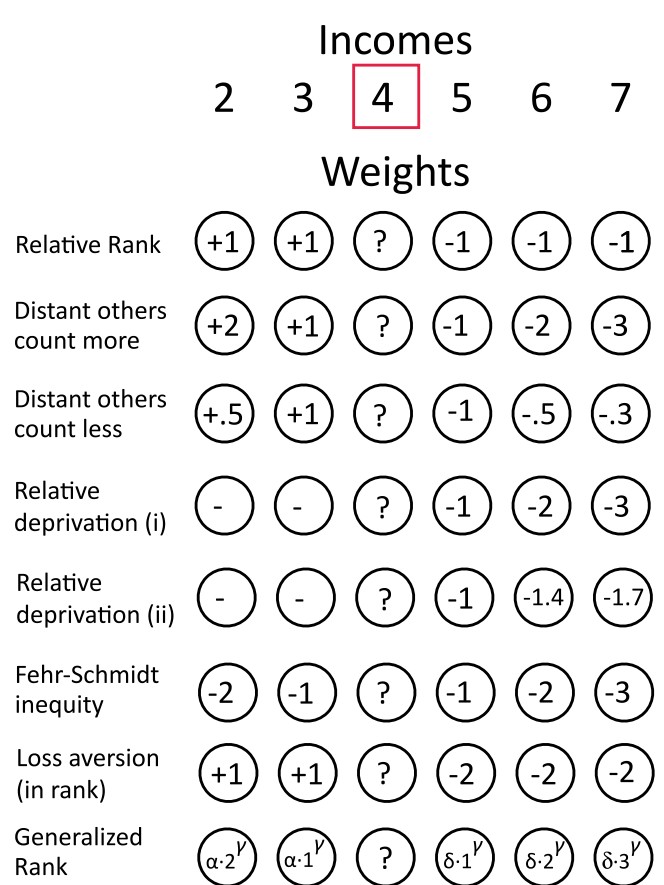

**Fig. 1 | Illustration of different weightings of comparison incomes on well-being.** The income to be evaluated is 4, in the context of comparison incomes 2, 3, 5, 6, and 7. First row: Pure rank account. Second row: Larger income disparities carry more weight than smaller ones; individuals who earn substantially more or less than *x* influence *x*'s well-being more than those who earn only slightly more or less than *x*[65]. Third row: People with incomes significantly different from *x*'s have less influence on *x*'s well-being than those with closer incomes, consistent with the idea that people often compare themselves to similar others (e.g., ref.[66]). Fourth row: Relative income deprivation, where only higher incomes reduce well-being, with weight increasing linearly with the income gap. Fifth row: Nonlinear deprivation, as in the Yitzhaki index[16] and its variations, where the influence of higher incomes grows in a curvilinear way with the gap. Sixth row: The inequity aversion model, where an individual's sense of fairness is affected negatively by both the total weight of incomes higher and lower than a certain income[67]. Seventh row: Loss aversion, where higher incomes reduce well-being more strongly than lower incomes improve it[68]. Eighth row: The generalized rank model used in this study, which flexibly weights upward and downward income differences and includes all prior models as special cases.

economic relative deprivation, in contrast, typically take into account the entire weight of income above that person's own income —the deprivation of a person earning \$20,000 will be increased more by a comparator earning \$50,000 than by one earning \$30,000. Although different measures of relative economic deprivation have been proposed (e.g., refs. [16,17]), they (unlike rank approaches) typically assume that a person's relative deprivation will increase as the sum of higher incomes earned by other people increases, even if the number of higher incomes is held constant. Economic measures of relative deprivation differ from measures developed within social psychology and sociology, stemming from Stouffer et al.[18,19], which incorporate subjective components (see, e.g., ref. [20]).

While socioeconomic gradients in health and well-being have long been reported[21], to our knowledge, no studies have directly contrasted rank-based and deprivation-based models of the association of

economic status with either SWB or health across multiple countries, although some single-country comparisons have recently been reported[5,17,22,23]. An additional issue concerns SWB's association with relative, rather than absolute, income. Almost all existing evidence linking SWB to ranked, rather than absolute, income comes from single-country studies, where the two measures are highly correlated (mean correlation of 0.97, across the 109 countries in the present study). This collinearity impedes interpretation (see ref. [24]). Cross-country studies that have compared absolute income either with the within-country rank of income[10] or with the within-country mean-relative income (e.g., ref. [25]) typically rely on country-level income measures as predictors (e.g., fixed effects), effectively reintroducing high collinearity and removing the variation needed to distinguish between the two. Finally, concern with income-based social rank is plausibly higher in more marketized/neoliberal economies and lower where social capital is higher[26,27], yet these potential moderating effects—which may constrain psychological interpretation—have received almost no attention.

Here, we outline and calibrate a general model to compare deprivation and relative rank accounts of the income-SWB relationship. The model includes absolute income, income rank, and several relative income deprivation models as specific cases, allowing for direct statistical comparisons within a unified framework (see also refs. [5,17,22,23]).

Figure 1 illustrates several ways in which an individual's perception of their income could be influenced by other incomes within a social comparison set. In "Methods," we formalize the general model and show that all the possibilities illustrated in the figure can be seen as special cases.

The model has three key parameters. Parameters $\delta$ and $\alpha$ determine the weight given to upward and downward income comparisons, respectively, with one typically fixed at 1. A parameter $\gamma$ governs sensitivity to distance: when $\gamma < 0$, individuals prioritize incomes closer to their own; conversely, when $\gamma > 0$, more distant incomes carry more weight. When $\gamma = 0$, all incomes above or below the individual's are equally weighted (i.e., only income rank matters). This approach enables nested model comparison. For instance, setting $\gamma$ to 0 yields a rank-only model whose fit can be compared to one with a free $\gamma$. A significantly better fit for the latter would suggest that income distances matter. Similar tests can be conducted to assess asymmetries in sensitivity to upward versus downward comparisons (i.e., loss aversion).

We therefore structure our model comparisons as follows. First, we test the *ubiquity* of income rank's association with SWB as compared to absolute income's association (see "Methods" for definitions). We find that income rank consistently outperforms absolute income in predicting four different measures of SWB in almost all countries we examined. The second set of analyses tests the *purity* of income rank's link to SWB. Specifically, using the general model described in Eq. (3) (Methods), we focus on distinguishing the income rank hypothesis from the relative deprivation income hypothesis. To foreshadow: we find strong evidence that social status (operationalized as relative income rank), not relative income deprivation, is independently associated with SWB. The third and fourth sets of analyses test the *robustness* of income rank associations by exploring moderators of the associations at both the country-level and individual-level.

We apply the model to data from 109 countries and more than 90,000 individuals, and examine both country-level and individual-level moderating variables, with a particular emphasis on social capital, market orientation, and materialistic values. Our outcome variable is SWB, which is correlated with objective measures of both health and economic well-being[28] and is increasingly seen as a target of public policy[29]. We focus on purely income-related measures of both social status and relative deprivation, as only by doing so can we compare social rank and relative deprivation accounts using nested model-based analysis.

## Results

### Analysis 1: income vs income rank

Our first analysis investigates, across many countries, whether income's relative rank or its absolute value is independently associated with life evaluation. To distinguish between the two, our empirical approach leverages cross-sectional variation between countries within survey years, using data from countries that vary widely in wealth. This method avoids the multicollinearity problem inherent in within-country analyses (see "Methods"), but raises concerns about unobserved heterogeneity across countries. To mitigate these concerns, we include a rich set of country-level variables capturing variation in economic development, social structures, and institutional context. We also extend the analysis to affective components of SWB, including positive affect (combining enjoyment and laughter) and negative affect (combining sadness, anger, worry, and stress).

The analysis uses the latest available GWP data collection round (2023–2024), which we replicate across earlier survey years in Supplementary Tables 3–6. Summary statistics (by GDP per capita quartiles) are shown in Supplementary Table 2. Life evaluation tends to be higher in countries with higher GDP per capita, with a two-point gap between the highest and lowest income quartiles on a 0–10 scale. Supplementary Fig. 1 shows the relationship between income measures and life evaluation, suggesting a linear relationship for income rank and a slightly concave one for log income.

First, we confirm that our approach successfully mitigates collinearity between absolute income and income rank. The correlation between absolute income and income rank is only $r = 0.50$ (95% CI [0.50, 0.51]) in the present study, as compared with within-country correlations averaging $r = 0.97$ ($SD = 0.04$, range [0.72, 1.00]). Thus, our results are unlikely to reflect multicollinearity.

Table 1 presents results from OLS regression models. Columns 1–4 show the effects of absolute income (measured as log income) and income rank separately, while column 5 combines them. Baseline specifications (columns 1 and 3) reveal significant positive associations for both income measures. In Column 1, the coefficient of 0.66 implies that moving from the 25th to the 75th percentile of income (approximately $4413 to $26,188) is associated with a 1.18-point increase in life evaluation on a 1–10 scale. Column 3 reports a coefficient of 1.39 for income rank, corresponding to the estimated difference in well-being between individuals at the very bottom and top of the national income distribution. This effect exceeds that of a substantial increase in absolute income. With controls, the absolute income coefficient drops by 50%, while the income rank coefficient decreases by 20%. In Column 5, absolute income becomes non-significant. The income rank coefficient in this fully specified model indicates that moving from the bottom to the top of the distribution is associated with a 0.97-point increase in life evaluation. A consistent pattern is found in earlier survey years (Supplementary Table 3), with the exception of 2017–2018.

Table 2 examines the association of income rank and absolute income with positive affect, negative affect, and predicted future life evaluation. In all three cases, income rank was the only significant predictor in the combined analyses.

To assess robustness, we tested the consistency of results across alternative specifications, including (i) alternative approaches to addressing missing covariate data, (ii) use of country fixed effects, (iii) gradual inclusion of individual- and country-level controls, and (iv) use of household-size equivalized incomes (see Supplementary Note 4). We also ran separate analyses for high- and low-income countries.

Most results closely mirrored those of the main specification, showing associations of income rank with SWB but not of absolute income. However, when country-level controls were excluded, the coefficient for absolute income became statistically significant, likely because absolute income partially captures country-level differences in infrastructure, services, and general development—factors that

extend beyond individual purchasing power, but which are captured by our inclusion of macro-level covariates. The robustness analyses also found that when income rank is based on equivalized income, it is no longer significantly associated with SWB. This result is consistent with the idea that social comparisons are based on visible, culturally salient indicators, such as unadjusted (gross) income, that align with how income is commonly discussed, displayed, and institutionalized, rather than on less visible metrics like household equivalence. Finally, the coefficient on income rank was larger for lower-income than for higher-income countries. However, consistent with the main analysis, only income rank, not absolute income, was independently associated with SWB in both country groups.

Overall, results show that income rank is more strongly and consistently associated with SWB than absolute income. When both measures are included, absolute income is not statistically significantly associated with SWB in most analyses. Other multi-country studies (e.g., ref. 25) have found only small effects of relative income, typically defined as income relative to the national mean. However, these estimates are difficult to interpret, as increases in mean-relative income, holding absolute income constant, may reflect not only shifts in relative standing but also broader changes in national income, potentially capturing other socioeconomic disparities.

Our coefficient on income rank for life evaluation is substantial. For example, it is twice as large as the coefficient on having a college degree and four times as large as the effect of being single rather than separated. It is also roughly equivalent to twice the difference in life evaluation between being unemployed and being employed full-time. Given these findings, we proceed to compare different specifications of relative income without further consideration of absolute income.

### Analysis 2: income rank vs relative deprivation

In our second analysis, we distinguish between the income rank and relative deprivation hypotheses and examine whether individuals place greater weight on upward income comparisons, consistent with loss aversion. We define income rank as an individual's ordinal position in the national income distribution, normalized between 0 and 1. A higher income rank indicates a better standing relative to others. Relative deprivation, in contrast, captures how much worse off someone is compared to those earning more. It takes into account not just whether others earn more, but how much more they earn.

We estimate life evaluation as a linear function of the generalized rank measure $R_i$ (see Eq. (4) in "Methods"), using maximum likelihood estimation (MLE) to recover the marginal effect of $R_i$ (i.e., $\beta$) and rank parameters ($\delta$ and $\gamma$), with $\alpha$ set to 1.

Simultaneously estimating $\delta$ and $\gamma$ can cause parameter instability. Therefore, to test for asymmetry in upward and downward comparisons, we fix $\gamma$ at 0 and vary $\delta$. When $\delta > 1$, upward comparisons weigh more, reflecting loss aversion. To assess sensitivity to income distance, we set $\delta$ to 1 and vary $\gamma$. When $\gamma < 0$, closer incomes carry more weight, but as $\gamma$ exceeds 1, more distant incomes are weighed more heavily, consistent with relative income deprivation.

Unlike Analysis 1, which pools countries, Analysis 2 estimates country-specific parameters to capture potentially heterogeneous social comparison processes. We begin with the test of comparison asymmetry. Supplementary Fig. 2 presents point estimates of parameters $\delta$ (loss aversion) and $\beta$ (marginal effect of income rank) using Round 18 of the GWP survey (2023–2024). Despite an average $\delta$ coefficient of 1.55, most $\delta$ estimates are not significantly different from 1.0 (55 out of 88 countries). Robustness checks using round 17 (2022–2023) revealed little consistency in $\delta$ estimates across years, suggesting limited evidence for asymmetric weighting (Supplementary Fig. 5).

Next, we report the key test of the purity of income rank associations, asking whether people compare their incomes primarily to those marginally better or worse off than themselves. The distribution

**Table 1 | Income effects on life evaluation, pooling countries**

| | Life evaluation | | | | |
|---|---|---|---|---|---|
| | (1) | (2) | (3) | (4) | (5) |
| Log income $1000 | 0.663 | 0.337 | | | 0.068 |
| | (0.570, 0.756) | (0.244, 0.430) | | | (−0.129, 0.266) |
| Income Rank Index (0–1) | | | 1.385 | 1.115 | 0.966 |
| | | | (1.239, 1.531) | (0.969, 1.260) | (0.493, 1.438) |
| Female | | −0.070 | | −0.024 | −0.028 |
| | | (−1.072, 0.932) | | (−1.018, 0.971) | (−1.026, 0.969) |
| *Education (base: elementary)* | | | | | |
| College degree | | 0.540 | | 0.494 | 0.489 |
| | | (0.387, 0.693) | | (0.328, 0.659) | (0.329, 0.649) |
| Secondary education | | 0.311 | | 0.292 | 0.289 |
| | | (0.166, 0.456) | | (0.138, 0.447) | (0.139, 0.440) |
| Unreported | | 0.453 | | 0.443 | 0.443 |
| | | (0.160, 0.746) | | (0.158, 0.728) | (0.156, 0.729) |
| *Employment status (base: employed full time)* | | | | | |
| Employed full time for self | | −0.020 | | −0.011 | −0.010 |
| | | (−0.117, 0.078) | | (−0.109, 0.087) | (−0.107, 0.088) |
| Employed part-time (do not want full-time) | | 0.136 | | 0.127 | 0.134 |
| | | (0.042, 0.230) | | (0.025, 0.229) | (0.040, 0.228) |
| Employed part-time (want full-time) | | −0.152 | | −0.172 | −0.161 |
| | | (−0.254, −0.050) | | (−0.285, −0.059) | (−0.263, −0.059) |
| Unemployed | | −0.593 | | −0.583 | −0.575 |
| | | (−0.718, −0.468) | | (−0.714, −0.452) | (−0.701, −0.450) |
| Out of Workforce | | −0.112 | | −0.084 | −0.083 |
| | | (−0.205, −0.019) | | (−0.182, 0.014) | (−0.180, 0.014) |
| *Health problems (base: yes)* | | | | | |
| No | | 0.557 | | 0.548 | 0.546 |
| | | (0.459, 0.656) | | (0.453, 0.644) | (0.451, 0.642) |
| Unreported | | 0.145 | | 0.156 | 0.157 |
| | | (−0.169, 0.459) | | (−0.157, 0.470) | (−0.157, 0.471) |
| *Marital status (base: single)* | | | | | |
| Married | | 0.024 | | 0.043 | 0.034 |
| | | (−0.082, 0.130) | | (−0.066, 0.153) | (−0.071, 0.140) |
| Separated | | −0.168 | | −0.186 | −0.182 |
| | | (−0.302, −0.034) | | (−0.322, −0.049) | (−0.319, −0.045) |
| Divorced | | −0.154 | | −0.112 | −0.116 |
| | | (−0.275, −0.032) | | (−0.232, 0.008) | (−0.234, 0.002) |
| Widowed | | −0.135 | | −0.095 | −0.100 |
| | | (−0.260, −0.010) | | (−0.221, 0.032) | (−0.223, 0.022) |
| Domestic partner | | 0.289 | | 0.234 | 0.241 |
| | | (0.166, 0.413) | | (0.114, 0.355) | (0.120, 0.362) |
| Unreported | | 0.258 | | 0.293 | 0.290 |
| | | (−0.155, 0.671) | | (−0.113, 0.700) | (−0.117, 0.697) |
| *Urban area (base: rural area)* | | | | | |
| Small town | | −0.009 | | −0.013 | −0.015 |
| | | (−0.149, 0.131) | | (−0.156, 0.129) | (−0.156, 0.127) |
| Large city | | −0.023 | | −0.005 | −0.013 |
| | | (−0.155, 0.110) | | (−0.141, 0.131) | (−0.144, 0.118) |
| Suburb of a large city | | 0.016 | | 0.034 | 0.026 |
| | | (−0.141, 0.174) | | (−0.126, 0.195) | (−0.130, 0.182) |
| Unreported | | −0.482 | | −0.439 | −0.449 |
| | | (−0.820, −0.144) | | (−0.772, −0.105) | (−0.784, −0.113) |

**Table 1 (continued) | Income effects on life evaluation, pooling countries**

| | Life evaluation | | | | |
|---|---|---|---|---|---|
| | (1) | (2) | (3) | (4) | (5) |
| *Country level controls* | | | | | |
| Ln health expenditure per capita $1000 | | 0.066 | | 0.295 | 0.250 |
| | | (−0.145, 0.277) | | (0.095, 0.496) | (−0.013, 0.514) |
| Unemployment (% of total labor force) | | −0.038 | | −0.046 | −0.044 |
| | | (−0.072, −0.004) | | (−0.079, −0.013) | (−0.078, −0.011) |
| Urban population (% of total population) | | 0.016 | | 0.017 | 0.017 |
| | | (0.005, 0.026) | | (0.006, 0.028) | (0.006, 0.027) |
| Gini Index (0–100) | | 0.002 | | −0.012 | −0.009 |
| | | (−0.023, 0.026) | | (−0.036, 0.013) | (−0.034, 0.016) |
| Absolute redistribution | | 0.003 | | 0.000 | 0.001 |
| | | (−0.021, 0.027) | | (−0.023, 0.023) | (−0.022, 0.024) |
| Constant | 4.216 | 6.540 | 5.053 | 6.998 | 6.840 |
| | (3.903, 4.529) | (4.991, 8.089) | (4.812, 5.295) | (5.478, 8.518) | (5.248, 8.433) |
| Observations | 97,339 | 97,339 | 97,339 | 97,339 | 97,339 |
| $R^2$ | 0.121 | 0.177 | 0.026 | 0.181 | 0.181 |

OLS models for the effect of income and income rank on life evaluation (for survey years 2023–2024). Columns 2, 4, and 5 incorporate individual-level controls, including age, gender (a four-degree polynomial of age and its interaction with gender), and country-level controls. Outliers below the 5th percentile and above the 95th percentile of incomes are excluded. Inference is based on two-sided $t$-tests with standard errors clustered at the country level; 95% confidence intervals are shown in parentheses. No multiple-comparison adjustments applied. Full regression results are provided in Supplementary Note 6.

of $\gamma$ and $\beta$ coefficients in Fig. 2 shows that $\gamma$ estimates for most countries are not significantly different from zero, consistent with the social-status income rank hypothesis rather than the relative income deprivation hypothesis. Estimates from adjacent survey years (Supplementary Figs. 6 and 7) show no consistency over time. Despite this, we further examined whether upward and downward comparisons reflect different reference groups by estimating separate distance parameters ($\gamma_u$ and $\gamma_d$), but the resulting estimates yielded similar null results (Supplementary Figs. 8 to 10).

Given the null results for asymmetry and distance sensitivity in income comparisons, we turn to our simplified income rank formulation ($\delta = 1$ and $\gamma = 0$) with $\beta$ as the only parameter. The distribution of $\beta$ coefficients (Supplementary Fig. 11) consistently shows positive and precise estimates of the income rank coefficient. The estimates also remain stable within countries over time, with a correlation of 0.75 across adjacent survey years and 0.39 over a decade (Fig. 3).

Finally, we present BIC scores to determine which income rank specification best fits each country's data. Supplementary Table 12 shows that the simple rank-based model, where each higher and lower income contributes equally to SWB, provides the best fit for 81 of 88 countries in Round 17 (2022–2023) and 79 of 88 countries in Round 18 (2023–2024) of the GWP. These results are consistent with the claim that people are sensitive to the relative ranked position of their income rather than to their relative income deprivation. Despite this, we observe considerable heterogeneity in the income rank coefficients across countries, raising the question of whether such country-level differences can be predicted. We examine this question in the next section.

**Analysis 3: country-level moderators**

Next, we explore potential country-level moderators of the income rank effect by incorporating interaction terms with each potential moderator into our models, while controlling for demographics and general country-level characteristics. We examine economic and institutional indicators (e.g., GDP per capita, competitiveness, openness, inequality, redistribution, unemployment, urbanization), aggregate population sentiment (e.g., personal security, confidence in institutions, community engagement, acceptance of migrants), and

societal values and economic preferences (e.g., individualism, materialism, risk/time preferences, altruism, reciprocity; see "Methods").

Figure 4 shows how the association between income rank and life evaluation varies with these contextual factors. The left plot shows marginal effects estimated at representative values (e.g., ±1 SD) of each moderator; the right plot shows differences relative to a baseline category (the first category within each group). Across all specifications, marginal effects are positive and significant, indicating a robust association between income rank and life evaluation across varying economic conditions, levels of institutional confidence, societal values, and economic preferences.

We first examine economic and labor market indicators. Income rank coefficients are smaller in countries where unemployment is low, possibly because income position is less salient in the absence of widespread economic insecurity. This pattern complements previous findings that SWB is more strongly associated with material income in poorer countries[25,30–33]. However, we find no consistent interactions with competitiveness or openness, and inequality does not moderate the association between income rank and SWB, consistent with Tan et al.[34] but differing from Macchia et al.[10]. We resolve this apparent discrepancy in Supplementary Note 5, where we reproduce Macchia et al.'s findings in their original 24-country sample. However, when we expand the analysis to 113 countries and extend the time frame to 15 years (2009–2024), the interaction between income rank and inequality becomes inconsistent across years, consistent with a limited overall role for inequality.

Turning to social capital, we find negative interactions with community involvement and willingness to volunteer and help others (Civic Engagement Index), and attitudes toward migrants (Migrant Acceptance Index). The magnitudes are significant; for example, when the Civic Engagement Index is increased from its lowest to highest value, the coefficient on income rank decreases by 80% (from 1.69 to 0.34).

Finally, we examine interaction effects involving societal values and economic preferences elicited under monetary incentives. We find larger income rank coefficients in societies with higher scores on the Materialist Index, which captures the extent to which respondents prioritize economic and physical

**Table 2 | Income effects on alternative well-being measures, pooling countries**

| | Life evaluation in 5 years | | | | |
| --- | --- | --- | --- | --- | --- |
| | (1) | (2) | (3) | (4) | (5) |
| Log income $1000 | 0.288 | 0.338 | | | 0.122 |
| | (0.183, 0.394) | (0.234, 0.441) | | | (−0.122, 0.366) |
| Income Rank Index (0–1) | | | 1.564 | 1.042 | 0.778 |
| | | | (1.388, 1.741) | (0.862, 1.222) | (0.166, 1.390) |
| Ln health expenditure per capita $1000 | | −0.309 | | −0.079 | −0.160 |
| | | (−0.570, −0.048) | | (−0.324, 0.165) | (−0.503, 0.183) |
| Unemployment (% of total labor force) | | −0.030 | | −0.038 | −0.035 |
| | | (−0.067, 0.007) | | (−0.073, −0.002) | (−0.073, 0.003) |
| Urban population (% of total population) | | 0.023 | | 0.024 | 0.024 |
| | | (0.011, 0.036) | | (0.011, 0.037) | (0.011, 0.036) |
| Gini Index (0–100) | | 0.023 | | 0.010 | 0.015 |
| | | (−0.003, 0.049) | | (−0.017, 0.037) | (−0.013, 0.042) |
| Absolute redistribution | | 0.003 | | 0.000 | 0.001 |
| | | (−0.023, 0.029) | | (−0.026, 0.026) | (−0.025, 0.027) |
| Constant | 6.350 | 5.669 | 6.244 | 6.181 | 5.900 |
| | (6.004, 6.695) | (3.884, 7.455) | (6.022, 6.465) | (4.441, 7.920) | (3.993, 7.808) |
| Individual-level controls | NO | YES | NO | YES | YES |
| Observations | 90,952 | 90,952 | 90,952 | 90,952 | 90,952 |
| $R^2$ | 0.023 | 0.135 | 0.032 | 0.137 | 0.137 |

| | Positive affect | | | | |
| --- | --- | --- | --- | --- | --- |
| | (1) | (2) | (3) | (4) | (5) |
| Log income $1000 | 0.019 | 0.020 | | | −0.025 |
| | (0.006, 0.031) | (0.001, 0.039) | | | (−0.071, 0.022) |
| Income Rank Index (0–1) | | | 0.143 | 0.107 | 0.161 |
| | | | (0.127, 0.160) | (0.091, 0.123) | (0.055, 0.267) |
| Ln health expenditure per capita $1000 | | 0.003 | | 0.017 | 0.033 |
| | | (−0.029, 0.034) | | (−0.009, 0.044) | (−0.014, 0.080) |
| Unemployment (% of total labor force) | | −0.004 | | −0.005 | −0.005 |
| | | (−0.010, 0.002) | | (−0.011, 0.001) | (−0.011, 0.001) |
| Urban population (% of total population) | | 0.001 | | 0.001 | 0.001 |
| | | (−0.001, 0.002) | | (−0.001, 0.002) | (−0.001, 0.002) |
| Gini Index (0–100) | | 0.005 | | 0.004 | 0.003 |
| | | (0.002, 0.008) | | (0.001, 0.007) | (−0.001, 0.006) |
| Absolute redistribution | | −0.002 | | −0.002 | −0.003 |
| | | (−0.006, 0.001) | | (−0.006, 0.001) | (−0.006, 0.001) |
| Constant | 0.668 | 0.659 | 0.639 | 0.652 | 0.708 |
| | (0.628, 0.708) | (0.436, 0.881) | (0.614, 0.664) | (0.454, 0.849) | (0.472, 0.945) |
| Individual-level controls | NO | YES | NO | YES | YES |
| Observations | 96,439 | 96,439 | 96,439 | 96,439 | 96,439 |
| $R^2$ | 0.004 | 0.048 | 0.011 | 0.052 | 0.053 |

| | Negative affect | | | | |
| --- | --- | --- | --- | --- | --- |
| | (1) | (2) | (3) | (4) | (5) |
| Log income $1000 | −0.038 | −0.026 | | | −0.006 |
| | (−0.047, −0.028) | (−0.037, −0.016) | | | (−0.031, 0.019) |
| Income Rank Index (0–1) | | | −0.124 | −0.086 | −0.073 |
| | | | (−0.140, −0.108) | (−0.099, −0.073) | (−0.133, −0.014) |

**Table 2 (continued) | Income effects on alternative well-being measures, pooling countries**

| | | | | | |
|---|---|---|---|---|---|
| Ln health expenditure per capita $1000 | 0.003 | | −0.015 | −0.011 | |
| | (−0.019, 0.025) | | (−0.036, 0.006) | (−0.040, 0.018) | |
| Unemployment (% of total labor force) | 0.000 | | 0.001 | 0.001 | |
| | (−0.003, 0.004) | | (−0.003, 0.005) | (−0.003, 0.005) | |
| Urban population (% of total population) | 0.001 | | 0.001 | 0.001 | |
| | (−0.000, 0.002) | | (−0.000, 0.002) | (−0.000, 0.002) | |
| Gini Index (0–100) | 0.001 | | 0.002 | 0.002 | |
| | (−0.002, 0.004) | | (−0.001, 0.005) | (−0.001, 0.005) | |
| Absolute redistribution | −0.000 | | 0.000 | 0.000 | |
| | (−0.003, 0.003) | | (−0.003, 0.003) | (−0.003, 0.003) | |
| Constant | 0.398 | −0.009 | 0.374 | −0.045 | −0.031 |
| | (0.373, 0.424) | (−0.192, 0.175) | (0.353, 0.394) | (−0.232, 0.142) | (−0.223, 0.160) |
| Individual-level controls | NO | YES | NO | YES | YES |
| Observations | 96,953 | 96,953 | 96,953 | 96,953 | 96,953 |
| $R^2$ | 0.022 | 0.085 | 0.012 | 0.086 | 0.086 |

OLS models for the effect of income and income rank on different individual well-being measures (for Survey Years 2023–2024). Columns 2, 4, and 5 control for age, gender (a four-degree polynomial of age and its interaction with gender), employment status, health problems, education, marital status, and urban/rural areas. Outliers below the 5th percentile and above the 95th percentile of incomes are excluded. Inference is based on two-sided $t$-tests with standard errors clustered at the country level; 95% confidence intervals are shown in parentheses. No multiple-comparison adjustments applied. Full regression results are provided in Supplementary Note 6.

security over self-expression and autonomy (reverse-coded Inglehart scale). The coefficients are also substantial. For example, when the Materialist Index increases from its minimum to maximum, the coefficient increases by 205% (from 0.61 to 1.86), more than tripling in magnitude. We do not find moderating effects of societal measures of risk-taking, patience, or reciprocity.

Detailed regression estimates of these marginal effects are found in Supplementary Tables 13–19. Supplementary Fig. 13 presents raw data patterns with fitted values for the 25th and 75th percentiles of the variables under analysis.

### Analysis 4: individual-level moderators

Finally, we explore individual-level moderators of the income rank effect, focusing on demographics, subjective sentiments such as personal security, institutional confidence, community life satisfaction, and economic preferences like risk and trust. These results parallel those from our earlier country-level analysis.

Figure 5 shows the marginal effects of income rank on well-being at representative values of individual-level indicators, revealing a consistent positive and significant association between income rank and SWB across all demographics, levels of confidence in society, and economic preferences. This pattern mirrors our country-level analyses.

Turning to specific moderators, we find smaller income rank coefficients for younger individuals (ages 15–28), single people, and part-time workers who would prefer full-time employment. With respect to social capital, the coefficient decreases as individuals report higher community life satisfaction, social support, community commitment, or trust in institutions (Community Index, Social Life Index, Civic Engagement Index, National Institutions Index). For example, the income-rank coefficient is 46% smaller at the maximum value of the Civic Engagement Index relative to its minimum (1.34 vs. 0.72). Among economic preferences, only risk-taking behavior interacts with income rank, with smaller coefficients for risk-takers. Specifically, the coefficient is 49% lower for individuals at the highest level of risk-taking compared to those at the lowest (1.42 vs. 0.73).

Overall, these results indicate that the income rank coefficient is smaller for individuals who have higher levels of civic engagement,

positive perceptions of social support from friends and relatives, and greater confidence in their government and key institutions. These variables mirror the values present in more communitarian societies with higher levels of social capital, where the importance of one's individual status within a social group can be superseded by the status of the social groups to which one belongs.

Detailed regression estimates supporting these marginal effects are provided in Supplementary Tables 20–22, along with raw data patterns in Supplementary Figs. 15 and 16.

## Discussion

We set out to answer three questions. First, is people's well-being linked to the relative ranked position of their income rather than to their relative income deprivation? We found clear evidence that it is: Pure income rank, rather than relative income deprivation, is independently associated with well-being in the large majority of countries and years, and for all four domains of SWB that we examined. Second, is income rank's association with SWB stronger than that of absolute income? Using a multi-country dataset that allows for a clearer separation between absolute and relative income than is possible in single-country studies, we found that it is. Third, are income rank associations smaller in countries with higher social capital, and larger in more open, marketized economies? We found that income rank coefficients are consistently lower in countries with higher social capital, but larger in countries with more materialistic cultural values. However, we observed no consistent interactions with economic openness or competitiveness. Income rank—not absolute income —was associated with SWB in both lower- and higher-income countries, though the coefficient was somewhat smaller in lower-income countries.

Our results appear consistent with the ideas that (a) social status influences well-being[14] and (b) relative rank of income is one, albeit not the only, index of such status. The results are also consistent with rank-based cognitive process-level accounts of how people make subjective judgments (e.g., the process-level accounts given by the Decision by Sampling model, refs. 15,35). According to such models, people sample quantities (e.g., incomes) from a population, and estimate the relative

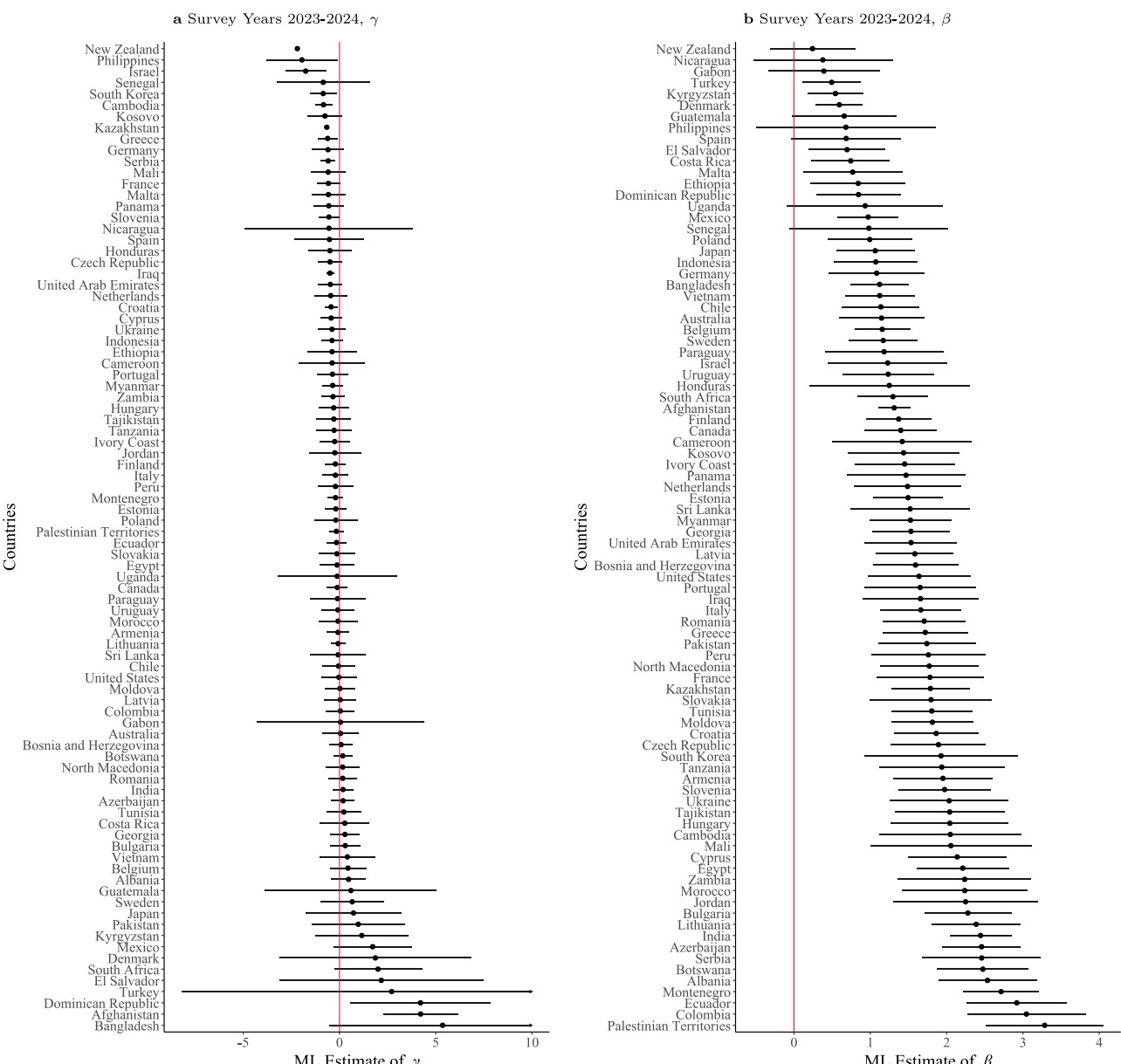

**Fig. 2 | Estimates of $\gamma$ and $\beta$ in a generalized rank model, allowing for dissimilar weights for distant Incomes. a** Maximum likelihood estimates of $\gamma$ in a model that weights equally upward and downward income comparisons by setting $\delta = 1$, $\alpha = 1$, and $-10 \leq \gamma \leq 10$. **b** Corresponding estimates of $\beta$. Points represent maximum-likelihood estimates for each country, based on nationally representative Gallup World Poll samples (typically about 1000–3000 respondents per country); lines denote 95% CI (omitted where variance estimation was numerically unstable).

ranked position of a given quantity, such as their own income, simply by counting up the number of sampled incomes that are higher and lower than the to-be-judged income (i.e., without taking into account of how much higher or lower the comparison incomes are). Rank-based sampling models of this type have received support from a large body of research on social comparison and judgment in other domains (e.g., mental health, ref. 36, alcohol consumption, ref. 37, and personality judgments, ref. 38).

Some limitations reflect the dataset. The GWP relies on self-reports of income, which may not be reliable[39], potentially attenuating the observed associations between absolute income and well-being. Any ineffectiveness in PPP adjustments may also add noise, militating against finding associations with cross-country components of absolute income. Only household, not individual, income is available, so our interpretation assumes that household income adequately proxies people's subjective income-related status. Our SES measures are estimates of objective, rather than subjective, socioeconomic status, and these distinct constructs relate differently to well-being[34,40]. Similarly, our measure of relative deprivation does not incorporate subjective components[19], and people systematically underestimate the relative ranked position of their income[41,42].

Moreover, our nation-based measure of rank excludes more local or interpersonal comparison groups[43]. If people are more likely to compare themselves to local rather than national peers when evaluating their socioeconomic status and well-being[40], this omission could lead to underestimation of the true effect of income rank.

While our analyses control for a wide range of individual-and country-level factors, the data remain observational, and our estimates should not be interpreted as establishing causal effects. In particular, the possibility of endogeneity—where the explanatory variable (e.g.,

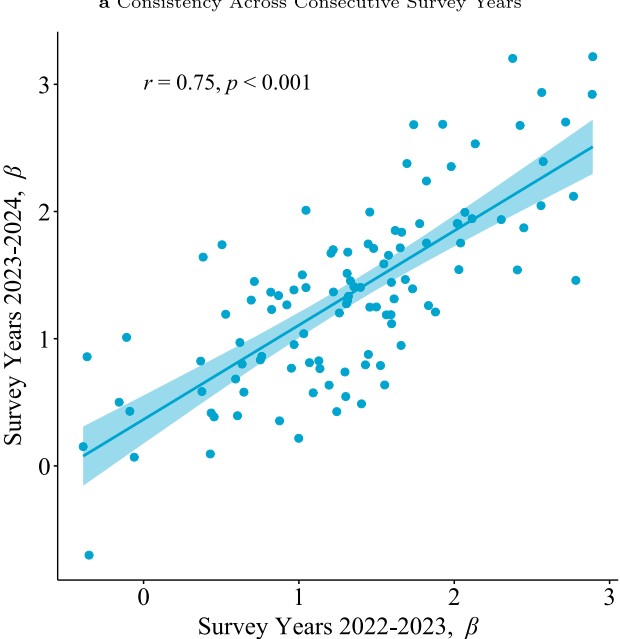

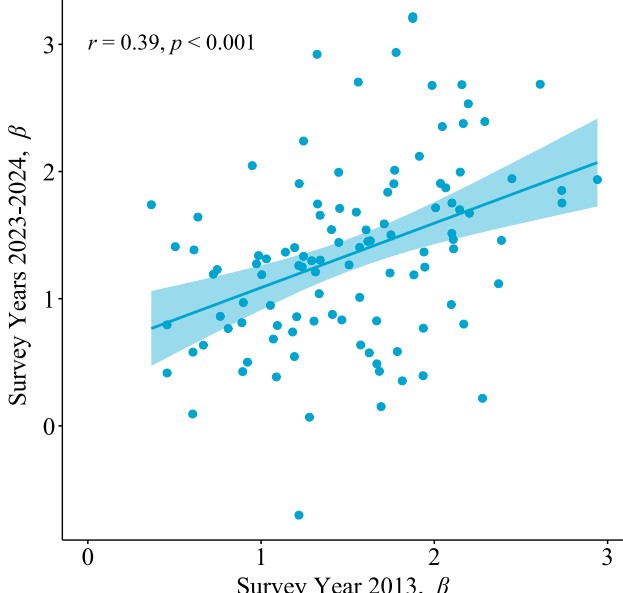

**Fig. 3 | Consistency of $\beta$ in a relative rank model. a** Scatter plot of country-level estimates of $\beta$, the marginal effect of income rank, across consecutive years. **b** Scatter plot of country-level estimates of $\beta$ after a decade. Both panels report estimates from a simplified model where $\delta = 1$, $\alpha = 1$, and $\gamma = 0$. Solid lines show the best-fitting linear relationships, and shaded areas denote 95% confidence intervals. Pearson correlation coefficients ($r$) and associated two-tailed $p$-values are shown in each panel. Plots include all countries for which the focal parameter could be estimated in both years. $n = 109$ countries.

income or income rank) is correlated with unobserved factors that influence well-being—limits our ability to make definitive causal claims. Endogeneity can arise from omitted variables, reverse causality (e.g., well-being influencing income), or measurement error. True causal inference would require exogenous variation in income, which our models do not provide. Future studies using experimental or longitudinal designs would be better suited to identifying causal pathways, although the use of panel data also brings its own problems, such as repeated-testing effects[44,45] and the possibility of unobserved trajectories[46].

Some moderation effects are particularly susceptible to endogeneity. Smaller income rank coefficients among individuals with higher trust or stronger social ties may reflect stable traits (e.g., optimism) that influence both social engagement and SWB. By contrast, national social capital likely captures broader institutional or cultural conditions that are less prone to individual-level bias. Moderation by national social capital is therefore less vulnerable to endogeneity and may more robustly reflect a buffering effect of social context on the psychological consequences of low income rank.

Although our results are not on their own informative about the direction of any possible causal relationship between income and SWB, because of the possibility of reverse causality[47,48], they are informative about the form of any such relationship. However, the three-way relationship between absolute income, income rank, and SWB may be different at different points of the relevant distributions (see refs. 2,49–52). Some studies have found that SWB has a bifactor structure, with a general factor as well as independent group factors[53–55], though such models are not straightforward to interpret[56] and could not be estimated here due to limited SWB indicators. Finally, we note that income is correlated with both consumption and wealth, both of which may predict SWB more strongly than income does[57,58].

The distinction between rank-based and relative deprivation accounts matters if SWB is a policy target[29]. Relative deprivation may be reduced through economic policy changes (e.g., reducing income inequality). To the extent that well-being is more strongly associated with income rank than with absolute income, in contrast, policies focused solely on raising aggregate income levels may have a limited impact on SWB. The implications for redistributive policies are more complex: While income rank is zero-sum, redistribution could affect well-being through other mechanisms, such as changes in inequality, social comparisons, or perceptions of fairness, which were not directly examined here.

## Methods

This research was confirmed as being exempt from ethical approval by the Leeds University Cross-Faculty Research Ethics Committee, Business, Environment, and Social Sciences.

We formalize a general model that encompasses all possibilities illustrated in Figure 1 as special cases[17,59].

The relative rank of $x$ is defined as:

$$R_i(x) = \frac{i-1}{N-1} \tag{1}$$

where $i$ is the ordinal position of $x$'s income in a comparison set of $N$ incomes.

Equation (1) can be rewritten as:

$$R_i(x) = 0.5 + \frac{(i-1)-(N-i)}{2(N-1)} \tag{2}$$

This form separates the number of incomes lower than $i$ (=$i$−1) and the number higher than $i$ (=$N$−$i$). To account for both the size and direction of income differences, not just their number, we extend this

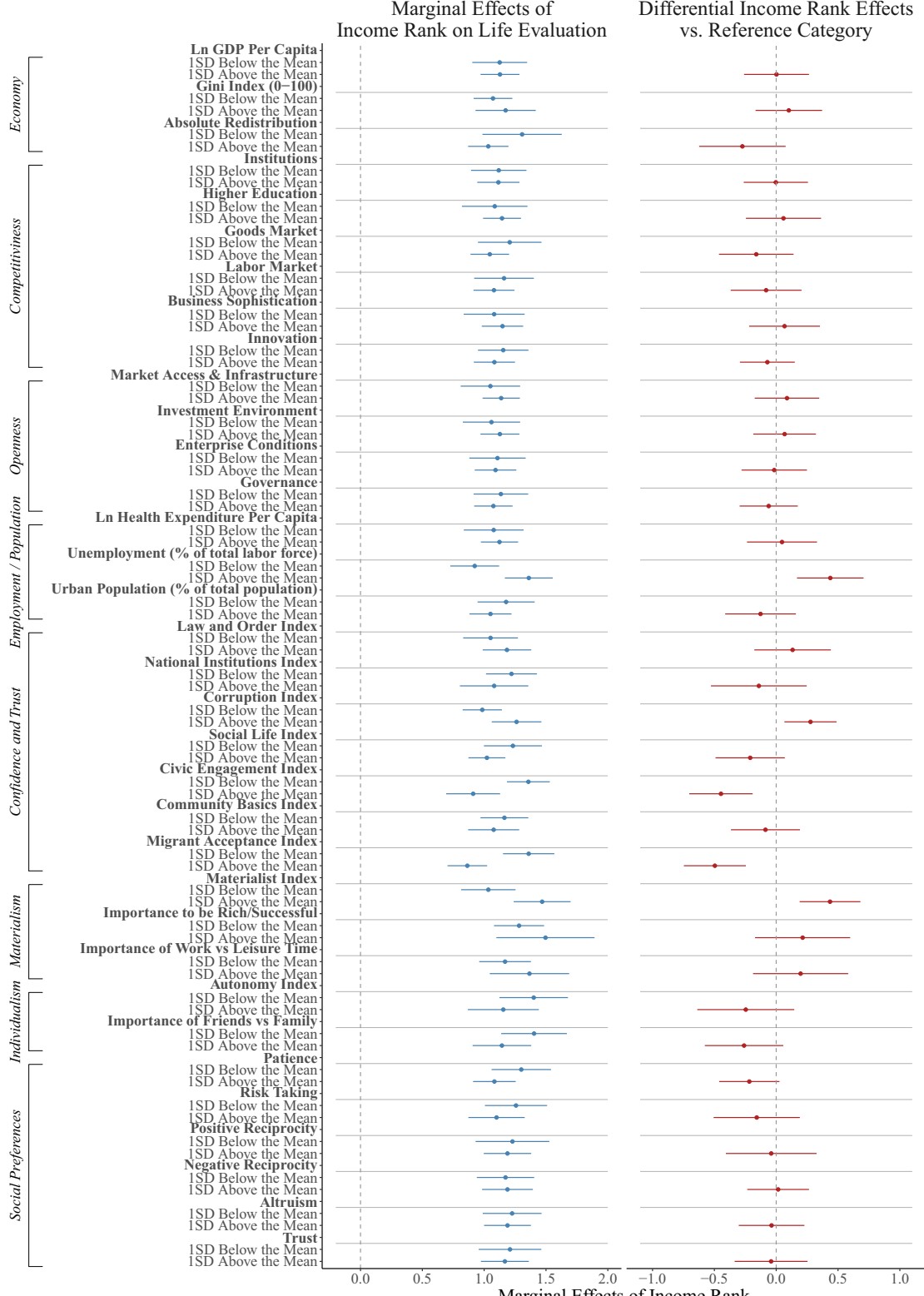

**Fig. 4 | Marginal effects of income rank across country-level indicators.** The left plot shows the marginal effects of income rank on life evaluation across different country-level characteristics. The right plot shows differences in effects relative to the reference category (first category in each group).

Points represent estimated marginal effects from independent regression models; horizontal lines indicate 95% confidence intervals with standard errors clustered at the country level. Corresponding sample sizes are reported in Supplementary Tables 13–19.

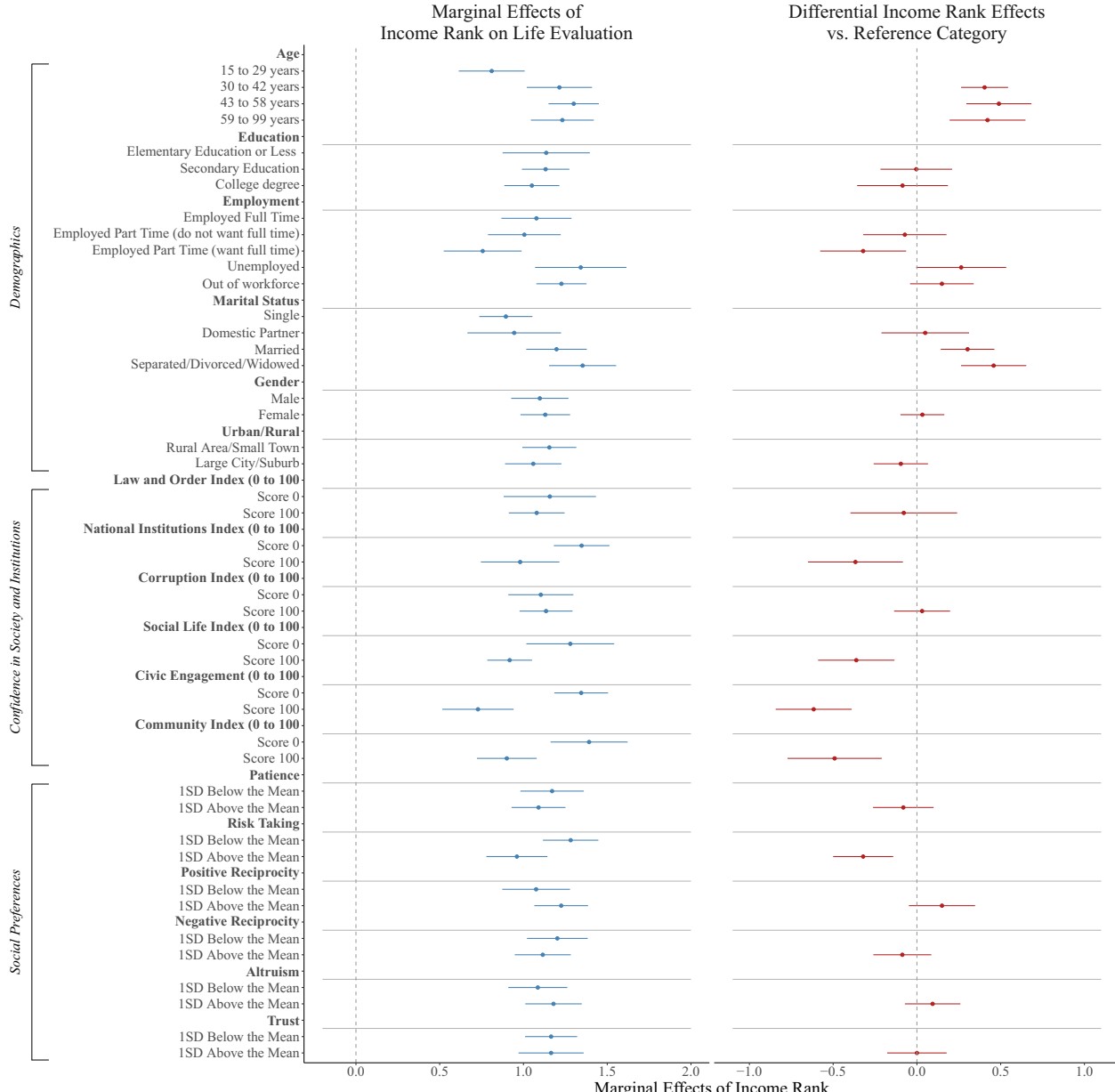

**Fig. 5 | Marginal effects of income rank across individual-level indicators.** The left plot shows the marginal effects of income rank on life evaluation across different individual-level characteristics. The right plot shows differences in effects relative to the reference category (first category in each group). Points represent estimated marginal effects from independent regression models; horizontal lines indicate 95% confidence intervals with standard errors clustered at the country level. Corresponding sample sizes are reported in Supplementary Tables 20–22.

formulation to include weighted comparisons:

$$R_i(x) = 0.5 + \frac{\alpha \sum_{j=1}^{i-1}(x_i - x_j)^\gamma - \delta \sum_{j=i+1}^{N}(x_j - x_i)^\gamma}{2\left(\alpha \sum_{j=1}^{i-1}(x_i - x_j)^\gamma + \delta \sum_{j=i+1}^{N}(x_j - x_i)^\gamma\right)} \qquad (3)$$

In Eq. (3), parameters $\delta$ and $\alpha$ determine the weight given to upward and downward income comparisons, respectively, with one typically fixed at 1. The parameter $\gamma$ governs sensitivity to distance: when $\gamma < 0$, individuals prioritize incomes closer to their own; conversely, when $\gamma > 0$, farther incomes carry more weight. When $\gamma = 0$, all incomes above or below the individual's are equally weighted (i.e., only income rank matters). Equation (3) simplifies to Eq. (1) when $\delta = \alpha = 1$ and $\gamma = 0$.

Equation (3) allows nested model comparison. For instance, setting $\gamma$ to 0 yields a rank-only model whose fit can be compared to one with a free $\gamma$. A significantly better fit for the latter would suggest that income distances matter. Similar tests can be conducted to assess asymmetries in sensitivity to upward versus downward comparisons.

To guide our empirical tests of income rank associations with SWB, we focus on three criteria: ubiquity, purity, and robustness. *Ubiquity* refers to the consistency of an association across contexts (e.g., countries and years); *purity* refers to the unique contribution of income rank beyond other income-related measures; and *robustness* refers to the need to deconfound absolute and relative income in cross-national studies as well as to avoid the incorrect assumption that relationships observed at the group level also hold at the individual level (i.e., the ecological fallacy).

Our analysis has four parts. We first test the ubiquity of the income rank hypothesis as an alternative to the absolute income hypothesis by analyzing its associations with four different measures of SWB. Next, we assess the purity of income rank associations with SWB. Using the general model described above, we distinguish the income rank hypothesis from the relative deprivation hypothesis by determining whether social status (relative income rank) or relative deprivation better accounts for SWB. In the third and fourth parts, we test the robustness of income rank associations by exploring potential moderators and by estimating country-level and individual-level moderation models separately, interpreting each within its appropriate scope. To avoid the ecological fallacy, we clearly distinguish between country-level and individual-level effects. Throughout the paper, we use the term "subjective well-being" as a generic term for self-reported mental state well-being, rather than any specific measure.

## Data

We use data from the GWP, analyzing six survey rounds covering a decade from 2013 to 2024: Rounds 8 (2013), 9 (2014–2015), 12 (2017–2018), 13 (2018–2019), 17 (2022–2023), and 18 (2023–2024). These rounds were selected to create three evenly-spaced pairs across the decade, allowing us to assess both short-term (consecutive rounds) and medium-term consistency of income ranks' associations with well-being. Although the GWP includes data from a larger number of countries, complete data across all six rounds, necessary for creating a balanced panel, were only available for these 109 countries. This approach helps rule out the possibility that our findings are driven by shifts in country coverage across survey years.

The original dataset includes 703,965 records from 109 countries. After removing entries missing well-being (life evaluation) or income data, the number reduced to 695,922 records. Further exclusions were made based on missing age information, bringing the count down to 694,290. Finally, we eliminated outliers in income (outside 5th and 95th percentiles), as per standard practice to avoid regression artefacts (see ref. 24), resulting in 608,226 observations for analysis. The GWP collects data through standardized interviews with informed consent from all participants. Respondents self-report their gender (male/female) and age, and gender was included as a covariate in all models. No statistical method was used to predetermine sample size. Randomization and blinding are not applicable to this observational study.

Our life evaluation metric is based on the Cantril Self-Anchoring Striving Scale. Additionally, we examined how income rank is associated with positive and negative emotions. For Analyses 3 and 4, individual-level moderators were sourced from the GWP, while country-level cultural, institutional, and economic moderators were derived from various other sources as described below.

- *Subjective Sentiment*. Using GWP indices as proxies, we assess societal attitudes towards various political, social, and economic topics.
- *Individualism and Materialism*. Drawing from the World Values Survey[60], we constructed measures of societal emphasis on autonomy, friendship, wealth, success, work, leisure, and post-materialist values.
- *Social Preferences*. The Global Preferences Survey[61] includes individual-level measures of time preference, risk preference, trust, altruism, and reciprocity, which we aggregated to form country-level indicators.
- *Economic Competitiveness and Openness*. We used indicators from the Global Competitiveness Report[62] and the Legatum Institute's Prosperity Index Report[63]. These metrics measure market competition, higher education, goods and labor market conditions, business sophistication, innovation, market access, investment environment, enterprise conditions, and governance.

- *General Country Characteristics*. We used the World Bank and Standardized World Income Inequality Database for data on GDP per capita, health expenditure, unemployment, urban population, and income inequality.

Our moderation analyses use data from GWP Round 18 (2023–2024), which includes both the SWB outcomes and access to the widest and most recent set of relevant moderators. This round also minimizes the risk of potential reverse causality, as all moderators are measured prior to the well-being outcomes. Supplementary Table 1 presents a correlation matrix for the country-level variables used in the paper. Further details on the country and individual-level variables are available in Supplementary Notes 2 and 3.

### Analysis 1: income vs income rank

Our first analysis investigates whether income's relative rank or its absolute value is more closely associated with life evaluation, extending previous research across a broad set of countries. As noted in the introduction, there is typically a very high correlation between an individual's income and their relative income rank within a single country, making it difficult to separate their associations statistically.

When analyzing multiple countries, prior research has typically employed country fixed effects to control for country-level differences and exploit within-country variation. However, this approach reintroduces collinearity. Country fixed effects remove all between-country variation in income, effectively centering income within each country. As a result, the coefficient on income no longer reflects individual-level material standards across countries but rather income relative to the national average. In this setup, absolute income becomes a within-country relative measure, conceptually similar to income rank. This conflation makes it difficult or impossible to meaningfully compare their independent associations.

Controlling for GDP per capita has a similar consequence: even without fixed effects, GDP per capita absorbs cross-country income differences, reducing absolute income to a residualized measure of relative standing (i.e., income relative to others within the same country). To address these limitations, we exploit cross-sectional variation across countries within each survey round, allowing us to retain both within- and between-country variation in income. This design avoids the collinearity issues inherent in fixed-effects models and enables a valid comparison between absolute and relative income coefficients. However, it raises concerns about unobserved country-level heterogeneity. To mitigate this, we include a rich set of macro-level controls: health expenditure per capita, the Gini index, the unemployment rate, and the percentage of the population living in rural areas. These variables help account for structural differences across countries that could otherwise confound the income-well-being relationship. We also cluster standard errors at the country level to account for within-country correlation.

To evaluate the association of SWB with income rank and absolute income, we estimate a utility function of the form shown in Eq. (4), where $U_{itc}$ is the utility of individual $i$ in country $c$ at year $t$ (proxied by their life evaluation); $x_{ict}$ is the natural log of individual $i$'s income, and **X** is a vector of control variables. These include demographic characteristics (a four-degree polynomial of age and its interaction with gender, as well as education, employment status, health problems, marital status, and urban/rural residence), and country-level characteristics (health expenditure, unemployment, urban population, Gini index, and absolute redistribution). Absolute redistribution is measured as the difference between market-income inequality (income before taxes and transfers) and net-income inequality (income after taxes and transfers).

$$U_{ict} = U(x_{ict}, R(x_{ict})) = \lambda_{ct} + \beta R_{ict} + \omega \mathbf{X}_{ict} + \epsilon_{ict} \qquad (4)$$

Income rank $R_i$ is calculated as the scaled position of $x_i$ in the ordered set of $N$ incomes (as in Eq. 2). For example, an individual with the 10th (20th) highest income out of 101 would have an index $R_i$ of 0.9 (0.8). Like most previous studies, we assume individual $i$ is sensitive to how their income ranks within the whole population of country $c$ in year $t$. Use of sub-national geographically or demographically defined comparison groups is not feasible given our cross-national approach and data availability. The GWP is designed to be nationally representative, but not necessarily representative within finer-grained subgroups. Constructing income rank within small subpopulations, such as by region, age, or education, would introduce considerable statistical noise and arbitrariness in group definitions. However, we note that the generalized model estimated in Analysis 2 allows for the differential influence of others who are more similar in the sense of having more similar incomes.

We estimate multivariate regressions with reported life evaluation as the primary dependent variable. All regressions were estimated treating life evaluation (0–10) as approximately continuous. Formal tests of linear model assumptions (normality and homoscedasticity) were not conducted, as estimates are robust in large samples. To mitigate concerns about reverse causality, we use country-level controls from the year prior to the survey, or from the most recent year available within the 5 years preceding the survey. We include dummies for missing controls, hence retaining the maximum number of observations.

As a robustness check, we repeated the analysis restricting the sample to countries with complete country- and individual-level covariate data. These results are reported in Supplementary Table 8.

Income is measured as annual household income in international dollars, calculated by the GWP using the World Bank's PPP estimates and expressed in 2017 US dollars. We focus on Round 18 (2023–2024) of the GWP for our main results, but replicate findings across survey rounds for robustness.

In addition, we extend our analysis to measures of affective SWB, examining both positive affect (a composite of enjoyment and laughter) and negative affect (a composite of anger, worry, sadness, and stress).

Finally, to further assess robustness, we conduct a series of supplementary analyses (see Supplementary Note 4) testing the consistency of results across alternative specifications. These include models with country fixed effects (Supplementary Table 7); incremental inclusion of individual- and country-level controls (Supplementary Table 9); use of household-size equivalized incomes (Supplementary Table 10); and separate analyses for high- and low-income countries (Supplementary Table 11).

## Analysis 2: income rank vs relative deprivation

In our second analysis, we distinguish between the income rank and relative deprivation hypotheses, and examine whether individuals place greater weight on upward income comparisons, consistent with loss aversion. As described earlier, income rank refers to an individual's ordinal position within the national income distribution. Relative deprivation, by contrast, captures how much worse off an individual is compared to higher earners, incorporating not just the number of people earning more but also the magnitude of those income gaps.

Our generalized income rank measure ($R_i$, ranging from 0 to 1), defined in Eq. (3), includes parameters $\alpha$, $\delta$, and $\gamma$ that enable flexible modeling of social comparisons. Parameters $\alpha$ and $\delta$ weight downward and upward income comparisons, respectively, enabling tests of loss aversion and income deprivation effects.

The parameter $\gamma$ captures distance sensitivity—whether individuals compare more strongly to those closer (negative $\gamma$) or further (positive $\gamma$) from their own income. When $\alpha = \delta = 1$ and $\gamma = 0$, Eq. (3) simplifies to Eq. (2), describing a standard relative rank measure where

equal weights are given to higher and lower earners in the social comparison.

To identify country differences in income comparison asymmetry, we use a simple univariate model, modeling life evaluations as linear functions of $R_i$ for each country, without individual controls. Unlike in Analysis 1, which pools countries, in Analysis 2, country-specific parameters are estimated to better capture potentially heterogeneous social comparison processes. Pooling could obscure these differences. We use MLE to estimate the intercept ($\lambda$), parameter $\beta$ (marginal effect of $R_i$, as in Eq. 4), and rank parameters ($\delta$ and $\gamma$), with $\alpha$ set to 1. This estimation does not rely on OLS assumptions such as normality or homoscedasticity.

Because estimating $\delta$ and $\gamma$ simultaneously can lead to instability, we examine each separately. To test upward and downward income comparison asymmetry, we fix $\gamma$ at 0 and allow $\delta$ to vary. When $\delta > 1$, upward comparisons weigh more, reflecting loss aversion. To evaluate distance sensitivity, we fix $\delta = 1$ and vary $\gamma$. When $\gamma < 0$, closer incomes carry more weight, but as $\gamma$ exceeds 1, further incomes are weighed more heavily, as suggested by the relative income deprivation hypothesis.

For computational efficiency and to avoid local maxima of the log-likelihood function, we bound our parameter space: $0 \leq \delta \leq 10$; $-10 \leq \gamma \leq 10$; $0 \leq \lambda \leq 10$; $0 \leq \beta \leq 10$. We discard out-of-bounds estimates, leaving 88 countries. Confidence intervals are computed from the Hessian matrix at the maximum likelihood estimates.

We first report results on asymmetry in upward and downward comparisons, followed by analyses of income distance sensitivity. Finally, we compare these generalized specifications with the simplified income rank formulation from Eq. (2).

## Analysis 3: country-level moderators

Analysis 3 explores potential country-level moderators of the income rank associations. We begin by investigating various measures of economic development and performance, including GDP per capita, inequality, economic competitiveness, and openness. Subsequently, we consider general socioeconomic factors such as unemployment and urbanization. We then shift our focus to aggregate population sentiments, examining aspects like personal security, confidence in key institutions, perceptions of corruption, social support, community commitment, community life satisfaction, and acceptance of migrants. Finally, we investigate societal values (e.g., individualism, materialism) and economic preferences, including risk and time preferences, trust, altruism, and reciprocity.

Moderation effects are assessed by interacting income rank with each moderator in Eq. (4). We analyze each moderator separately to prevent multicollinearity and potential bias in our estimates, as some of these candidate moderators (such as perceptions about personal security or confidence in institutions) could shape social preferences like altruism or reciprocity, or societal values like individualism or materialism. As in Analysis 1, we use linear regression, treating life evaluation (0–10) as approximately continuous and control for demographics and general country-level characteristics, including health expenditure per capita, unemployment rate, urban population rate, Gini index, and absolute redistribution level. All country-level variables, except the general characteristics, are standardized for comparability.

Due to data limitations, this analysis focuses exclusively on Round 18 (2023–2024) of the GWP. Many of the moderators of interest—such as economic competitiveness, openness, societal values, and incentivized economic preferences—are only available for recent years or are not consistently collected across survey rounds. Conducting moderation analyses across the full dataset is therefore infeasible. Using earlier SWB data would also risk reverse causality effects, as it would require pairing SWB outcomes with moderator variables measured in

later years (e.g., explaining 2014 well-being using 2017 competitiveness data). By restricting the analysis to Round 18, we ensure that moderators precede the well-being outcomes, reducing ambiguity about directionality.

To visualize moderation effects, we plot marginal effects of income rank at representative values of each moderator (e.g., ±1 SD). Detailed regression estimates used in calculating these marginal effects are provided in Supplementary Tables 13–19. For completeness, we also present (unconditional) raw data patterns in Supplementary Fig. 13, accompanied by fitted values on solid lines. Fitted values correspond to percentiles 25 and 75 of the variable under analysis while holding the remaining covariates as their mean values, or at their mode values for discrete covariates.

### Analysis 4: individual-level moderators

Analysis 4 explores individual-level moderators of the income rank coefficient. We begin by examining demographic differences. Next, we analyze individual subjective sentiment, covering the same aspects as in our previous country-level analysis: personal security, institutional confidence, corruption perceptions, social support, community commitment, and community life satisfaction. These social capital and subjective sentiment variables, calculated by the GWP, range from 0 to 100 and are not standardized, offering a clearer view of the income rank coefficients for individuals with extreme judgments. Finally, we investigate links between income rank coefficients and various economic preferences, such as risk, time, trust, altruism, and reciprocity, which are standardized across countries, as calculated by Falk et al.[61]

Following the same estimation procedure as for our country-level analysis, we analyze each potential moderator separately. We test the moderating role of each individual-level variable by adding an income rank interaction term to Eq. (4), while retaining the full set of control variables. We use data from Round 18 of the GWP for most analyses, except for the evaluation of economic preferences, whose individual scores are only available for Round 7 (2012–2013).

Detailed regression estimates used in calculating these marginal effects are provided in the Supplementary Tables 20–22. Here again, for completeness, we also present (unconditional) raw data patterns in Supplementary Figs. 15 and 16, accompanied by fitted values on solid lines. Fitted values correspond to percentiles 25 and 75 of the variable under analysis while holding the remaining covariates as their mean values, or at their modal values for discrete covariates.

### Reporting summary

Further information on research design is available in the Nature Portfolio Reporting Summary linked to this article.

## Data availability

The primary data supporting this study are available from the Gallup Organization (https://www.gallup.com/analytics/318923/world-poll-publicdatasets.aspx) to subscribed researchers or research advisors and may be made temporarily available under controlled conditions for peer review. Additional country-level cultural, institutional, and economic variables are drawn from publicly available sources: individualism and materialism from the World Values Survey; incentivized measures of economic and social preferences from the Global Preferences Survey[61]; economic competitiveness and openness from the Global Competitiveness Report[62] and the Legatum Institute's Prosperity Index Report[63]; income inequality from the Standardized World Income Inequality Database[64]; and GDP per capita, unemployment, and urban population from the World Bank's World Development Indicators, based on the latest available estimates at the time of download (April 2025), as detailed in Supplementary Note 2.

## Code availability

The study's analysis code is available at https://osf.io/bj9cz.

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

## Acknowledgements

This work received funding from the European Research Council (ERC) under the European Union's Horizon 2020 research and innovation program (grant no. 788826).

## Author contributions

E.Q.T. and G.D.A.B. conceived the research idea and designed the project. J.D.N. made the Gallup World Poll data available. E.Q.T. performed all analyses. E.Q.T. and G.D.A.B. co-wrote the paper, and J.D.N. revised it. All authors edited and approved the manuscript.

## Competing interests

The authors declare no competing interests.
