## [Transparent Peer Review file · Nature Communications]

Social status and the relationship between income rank and well-being in 109 nations

Corresponding Author: Dr Edika Quispe-Torreblanca

Version 0:

Reviewer comments:

Reviewer #1

(Remarks to the Author)

This is a review of the paper "Social Status, Relative Deprivation, and Social Capital: The Ubiquity, Purity and Robustness of Income Rank Effects on Well-being in 90 Nations." Using data from the Gallup World Poll, the article explores the role of income rank and income deprivation on wellbeing as well as several moderators and mechanisms. I will now proceed to comment on each section of the paper.

Point 1. Abstract

It would be good to know the dataset and methods used (e.g., what does 'we calibrate a general model' mean?). I would also be good to know how relative deprivation and income rank are calculated as readers who are not experts on these topics might not know what these two concepts are reflecting.

Point 2. Introduction

Point 2.1. SWB stands for 'Subjective wellbeing' not self-reported wellbeing as stated in the first paragraph of the introduction.

Point 2.2. The introduction is overall good with good detail about the studies.

Point 3. Results

Point 3.1. Study 1 should include sadness as another indicator of negative affect.

Point 3.2. Study 1. This result should be explained further "Although some panels indicate a negative association between absolute income and affect levels, this association is small and non-significant in earlier survey waves" why might this be the case?

Point 3.3. Study 1. It looks like the mediation results are based on adding positive and negative affect as additional covariates. Although this method can tell whether a variable can be a potential mediator, this cannot be called mediation analyses and some test like Structural Equation Modelling should be conducted.

Point 3.4. Study 2. Was the test in Study 2 also run with pooling countries as in Study 1?

Point 3.5. Study 3 and 4 are good and clearly explained although some concerns about waves still remain.

Point 4. Discussion

Point 4.1. Summary of results and limitations are well described. More on psychological mechanisms is needed.

Point 5. Methods

The article uses data from the Gallup World Poll (GWP) and there are a few points that need further clarification.

- Point 5.1. It is not clear why the data used is 2009-2019 if the GWP goes from 2005 to 2024 already.
- Point 5.2. A minor point is that the years might be better labelled as 'survey years' instead of 'waves' because 'waves' is usually used for longitudinal data and the GWP is cross-sectional.
- Point 5.3. The methods section says "Social Status, Relative Deprivation, and Social Capital: The Ubiquity, Purity and Robustness of Income Rank Effects on Well-being in 90 Nations." It is not clear how the authors came to this information because the GWP contains data from 165 countries and territories and more than 2 million respondents now.
- Point 5.4. The methods section states "Each wave was analyzed independently and compared with the immediately preceding wave for robustness checks." It is not clear why all survey years were not analysed together, and country and year fixed effects used in the regressions. If there is a clear rationale behind the choice of this method, this should be clearly explained.
- Point 5.6. Why the moderation analyses were conducted only with 2018-2019 survey years should also be explained. Is it because the relevant variables are not available the previous years? In the methods section it says that this choice was made to 'minimize the risk of potential reverse causality', why is this the case?
- Point 5.7. Study 1 replicated prior work with an alternative method. The authors challenged the method widely used in prior work by saying that this method introduced multicollinearity when also including GDP per capita as a control and country fixed effects. For example, the authors say that "When studying multiple countries, previous research has typically employed country fixed effects to control for country-level differences and exploit within-country variations. However, this method introduces multicollinearity due to the high correlation among income-related variables." If the issue is the within country-variation, the analyses should be conducted by each country not by each survey year or wave (I see an attempt of this in Study 2). This is because when conducting the analyses by survey year or wave, all the countries are still pooled together which is what the authors challenge in the method used in prior work. The findings presented in this paper should be replicated with this widely used method with and without GDP as a control.
- Point 5.8. It would be good to know how the mediation analyses were conducted. This is important because the dependent variable and the mediators are all self-reported and reverse causality, multicollinearity, and endogeneity might be present.
- Point 5.9. In section 4.3 about Study 2, income rank and relative deprivation need to be defined (e.g., what do these mean). We know how these were calculated from the equation details, but we don't know what they mean in plain words.
- Point 5.10. Study 3 is well explained but the use of only 'wave 13' should be better justified and robustness checks with all the survey years together conducted.
- Point 5.11. More detail is needed regarding each moderator used in Study 3 and Study 4.
- Point 5.12. Ubiquity, purity, and ecological fallacy should be defined in the first part of the methods section.

(Remarks on code availability)

I didn't revise the code because I had to request access to do so.

Reviewer #2

(Remarks to the Author)

The paper contributes to our understanding of the relationship between income and wellbeing, which is an actively debated topic in wellbeing studies. The authors first conclude that it is relative income within a given country, rather than the absolute level of income, that matters for wellbeing. They then explore various approaches to measuring relative income, finding that a simple rank-based measure best fits the data. Finally, the authors demonstrate how the relationship between income rank and wellbeing varies across countries and among individuals with different characteristics.

It is a important contribution that the simple measure of rank performs as well, at least globally, as more complex measures accounting for asymmetries (Study 2). Your heterogeneity analyses (Studies 3 and 4) offer almost encyclopaedic detail — I find the results on civic engagement particularly interesting. However, I believe the main finding of the paper comes from Study 1, where you conclude that relative income rank within a country matters more for wellbeing than absolute income levels. I would want to better understand how one should interpret this result in the context of your modelling approach.

1. The level-of-income vs. relative-income result is derived from a pooled cross-country regression without country fixed effects. The model controls for household income and income rank at the same time. This model specification is unusual. A standard cross-country wellbeing model would include household income and country fixed effects. The coefficient on household income from a standard model is then interpreted as the relations between wellbeing and higher or lower income in a given country, which is effectively relative income. The authors remove country fixed effect and include income both as a level (which is not significant) and as a percentile of income distribution (which is significant). The latter, effectively, captures similar variation to the coefficient on income from a model with county fixed effect. As a results, though the model formulation is unusual, I do not find the result surprising.
2. In Table 1, R-squared in Model 1 (with income in levels) is three times higher than in Model 3 (with relative income), suggesting that a combination of income level and wellbeing fits the data better than a combination of income rank and wellbeing. It would be helpful to see a model that includes both income variables without additional controls. I would expect the coefficient on log income in this model to be positive, because having higher income conditional on a given income rank implies living in a wealthier country. If it is not, it would be helpful if you could provide some intuition on why that is the case.
3. I assume that the coefficient on individual income in Model 5 is not significant, because the wellbeing benefits of leaving in a richer country are captured by controlling for health expenditures (correlation with GDP of 0.96) and absolute redistribution (correlation with GDP of 0.76). It is not clear to me how to select a set of country-level controls that allow controlling for heterogeneity without controlling away the effects of leaving in a wealthier country. It would be helpful to see the sensitivity of the results in Column 5 of Table 1 to including subsets of controls, e.g. individual level only, country-level with low correlation with GDP, etc.

4. The authors state that the magnitude of the coefficient on the income rank is high. One can interpret the magnitude of this coefficient as a wellbeing difference between the poorest and the richest household in a country. Then, the coefficient of 1.5 on a 0 to 10 scale in Column 5 of Table 1, though still large, does not look enormous. It would be helpful to see some intuition on how to compare relative magnitudes of the coefficients on income level and income rank, e.g. the expected effect of moving from the 25th to the 75th percentile according to the estimates from Columns 1 and 3.

5. I don't believe that your findings corroborate the following statement in the conclusion: "The zero-sum property implies that, to the extent that a given domain of SWB is affected by income rank rather than absolute income, neither general increases in societal income nor redistributive policies will, in themselves, increase SWB." Firstly, given that your model effectively controls for GDP per capita, which can be interpreted as the level of societal income, I believe that non-significant coefficient on individual income does not prove that general increases in societal income will not increase wellbeing. On redistributive policies, given that your analysis is cross-sectional, it is not informative about (potentially asymmetric) wellbeing benefits and losses from gaining or losing rank for people in different parts of the distribution. Since your research does not speak to that, I don't think that it can be used to support a statement about potential (non-)effectiveness of redistributive policies.

Minor comments:

1. Wellbeing literature that uses comparator group typically defines it as a group with similar characteristics (e.g. age, gender, education, region). Could you please discuss why you believe national comparison to be the best choice in your case. Especially given that using subgroups of the population might also help you to alleviate concerns of within-country multicollinearity between income level and income rank.

2. I am concerned about negative coefficient on the level of income in models for positive affects will all controls, both in the main result (Table 2) and in wave 12 (Table S5). Could you please provide some intuition on why this might be the case?

3. It would be helpful to see if your main results in Table 1 hold for the equivalence-scale adjusted household income, since it is more accurately represents the level of income individual enjoys.

Thank you for your work.

(Remarks on code availability)

I could not access the codes because the files linked above, https://osf.io/bj9cz/?view_only=8e0e7ff929e34cdab499695ef5843796, require submitting a request for permission to gain access.

Reviewer #3

(Remarks to the Author)

Comments on Social Status, Relative Deprivation, and Social Capital: The Ubiquity, Purity and Robustness of Income Rank Effects on Well-being in 90 Nations
Submitted to Nature Communications
NCOMMS-24-42657-T

This paper studies the relationships between individuals' absolute income and relative income and their subjective wellbeing. We know that wellbeing increases with income (and that the effect is in part causal) but it is an ongoing topic of research as to how much is due to absolute vs relative effects. The absolute component is the degree to which an individual has more resources at their disposal. The relative component reflects the fact that a higher income might imply a greater position within a social hierarchy. If relative effects are dominant, this has some troubling implications for society, as it implies that economic growth (rising absolute incomes everywhere) may be limited in their ability to produce greater wellbeing.

The research here is an extremely comprehensive empirical exercise looking at the associations between income and income position (and a couple of related/derived measures) with wellbeing across 90 countries. The main innovation comes from the breadth of data used, and the recognition that country fixed-effects (normally a positive feature of these models) is actually a liability within this context. The reason is that an individuals' income and their income rank (which in turn capture the absolute and relative components) are very highly correlated (the authors quote a correlation coefficient of 0.98). This means that it is immensely difficult to disentangle the effects of each, as there aren't many examples in the data where an individual's absolute income is much higher or lower than what would be implied by their rank.

The pooling of many country-level datasets offers a partial solution as ranks get defined within countries while income gets a singular definition across countries. So by comparing people with \$X in poor countries and in rich countries (where the absolute value is constant, but the relative position changes) the authors can resolve this problem. The note that this pooling introduces a lot of heterogeneity, which they resolve with an extensive set of controls. The authors find considerable support for income ranks as a dominant driver of wellbeing, and distinguish this from a related concept of relative deprivation. They also find that income ranks are less important in countries with high levels of civic engagement, and uncover suggestive evidence that countries with more neoliberal politics might place more emphasis on relative effects.

This is a very strong paper. A high level of technical and conceptual expertise is shown throughout. The writing is careful, clear, and well-structured. I believe the paper is suitable for Nature Communications but have made some comments and suggestions below. All should be taken seriously, but I don't feel any are particularly critical.

- The analysis hinges very strongly on how well the country-level controls remove heterogeneity that could be correlated with income and may potentially explain wellbeing. While we can't tell for sure whether or not the controls are sufficient, if the results are heavily dependent upon a particular control, this may imply that that underlying factor isn't properly adjusted for. The authors could examine this further. It is a bit difficult to see what is happening as they just report "individual controls – yes/no" (and the same for countries). And sometimes the results bounce around and sometimes they don't.
- The authors don't use the concepts of exogeneity/endogeneity at all in the paper. I appreciate that this is a general interest

journal, and the readership may not be versed in econometric terminology, but these ideas are too useful to just gloss over. Since the pdf is 75 pages long it would be easy to briefly explain these concepts and make it clear to the reader that the causal claims are fairly muted.

- Following on from the above point, the authors should probably further mute their causal claims and tone down the causal language.
- I have my doubts about using the indicator approach used to handle missing data. This is a simple method where a missing value for a covariate is imputed (usually with the mean, but it doesn't matter), and an additional dummy variable is included in the regression to indicate that a value was imputed. This is only done for the country-level data, but it can be a source of bias in non-experimental studies (e.g., Groenwold et al., 2012). Does this improve statistical power? Maybe, because you get more data, but maybe not, because you have to estimate more covariates. Why not use a standard multiple imputation technique? Or just report results for the data you have?
- Out of interest, what happens to the estimates in Table 1 when the fixed-effects are included?
- It is tempting to think that relative effects will become larger in richer countries, which the authors demonstrate. But do absolute effects dominate in poorer countries? Surely a starving child cares much more about food than they do about social status. This would be simple to carry out and could be reported in an appendix. If true, it would also serve as a rough intuition check for the rest of the paper.
- Some of the tables are too big and contain too much information to be easily interpretable.
- The fact that income ranks seem less important in countries with more social capital, and for individuals with more social capital, seems like a meaningful and under-discussed result. It is easy to see how the concept of endogeneity could be used to explain the latter (e.g., people with positive social characteristics may be autonomously happy and insensitive to income in general) but not the former.
- The authors worry a little that their self-reported income data may not be reliable. If miserable people over-report their income (which seems more likely than the reverse) then this would attenuate the results. So the real correlations might be stronger than the ones reported here.
- The depressing conclusion that income rank is zero sum could be expanded upon. The stakes of this research are quite high, and a better discussion of the unpleasant result and its social implications would help.

(Remarks on code availability)

I had a quick look at the code. It looks very professional, but I didn't scrutinize it for errors. Since the data is public it should be straightforward to reproduce all results.

Reviewer #4

(Remarks to the Author)

This is a very ambitious paper. I think the paper's most significant and interesting contribution is to tease apart the differing effects of income rank and relative deprivation--showing that income rank is by far a stronger measure (Study 2). The exploration of heterogeneous effects of income rank in Studies 3 and 4 is interesting as well and complements Study 2 nicely.

My largest concern is with study 1, which examines absolute vs relative income. The authors note the problem that "Within single countries, incomes and relative income ranks are highly correlated, making it difficult to separate their effects statistically." (p. 6). It seems the problem is "solved" by pooling countries, so effectively our variation is by comparing two people with the same income, but one in a poor country where they have a higher rank) to someone in a rich country (where they have a lower rank).

But, the authors explain in the methods that "We measure income as annual household income in international dollars, calculated by the GWP using World Bank's PPP estimates, represented in 2011 US dollars." So, in my example above, two people with the same income (as measured in international dollars) have the same purchasing power, regardless of which country they are in. Thus, it's little surprise that their income rank would have more influence than absolute income.

Also related to Study 1, in the conclusion of the paper, the authors finish by stating that "...The zero-sum property implies that, to the extent that a given domain of SWB is affected by income rank rather than absolute income, neither general increases in societal income nor redistributive policies will, in themselves, increase SWB." (p. 19). But, note that the models from Study 1 have included measures such as log health expenditures per capita (which has a large positive effect on SWB, Table 1), and which has a .96 correlation with log GDP (Table S1). If a general increase in societal income is part of an increase in GDP or health expenditures (or whatever else these proxy for), then this likely does increase SWB.

Other (more minor) comments:

* Social capital is in the title, but it plays a very minor role in the results (alongside with other potential mediators), and has no part in the background. There is no review of the extensive literature on how social capital is measured and how it may affect SWB. I think these are still fine analyses to have, but "social capital" certainly shouldn't be part of the title, as it's not what the paper is about.

* In study 1, the authors write: "These results show that (a) the rank of a person's income is a better predictor..." As far as prediction accuracy, the R-squared of both rank and absolute income are similar (and both are low). The authors mean something like a larger effect (although the units are different...)--regardless of how they want to say it, more care is needed here.

* I like the moderation analyses in Studies 3 and 4, but the figures fall a bit short in conveying the key information. The evidence of an effect is the size of the gap between the two predicted marginal effects for each variable. This is hard to eyeball. I wonder if the authors could find a way to include this information as well?

* In the supplement, the authors write "In our study design, we do not conduct a power analysis to determine our sample size, as we are not powering for null hypothesis significance testing." But really, as soon as you are reporting statistical significance tests, that's exactly what you are doing. The statistical significance test is based on the logic of null hypothesis testing. I don't necessarily feel that power analyses are needed, but this statement should be omitted at the very least.

(Remarks on code availability)

I looked at some of the code (including the README), but the data are not available, so I did not run anything.

Version 1:

Reviewer comments:

Reviewer #1

(Remarks to the Author)

The authors provided comprehensive responses to the comments. The only issue is that I couldn't still review the code. As far as I could see the file included in the OSF link contains a list of R scripts used but not the actual scripts. Actual scripts should be readily accessible in the OSF project.

(Remarks on code availability)

The file included in the OSF link contains a list of R scripts used but not the actual scripts. Actual scripts should be readily accessible in the OSF project.

Reviewer #2

(Remarks to the Author)

Dear Authors,

Thank you for addressing my comments from the previous round and for the additional analyses. I am satisfied with your responses, and I have no further comments at this stage.

Congratulations on your work.

(Remarks on code availability)

Reviewer #3

(Remarks to the Author)

This is a solid revision. I have no further comments.

(Remarks on code availability)

REVIEWER COMMENTS

Reviewer #1 (Remarks to the Author):

This is a review of the paper “Social Status, Relative Deprivation, and Social Capital: The Ubiquity, Purity and Robustness of Income Rank Effects on Well-being in 90 Nations.” Using data from the Gallup World Poll, the article explores the role of income rank and income deprivation on wellbeing as well as several moderators and mechanisms. I will now proceed to comment on each section of the paper.

Point 1. Abstract

It would be good to know the dataset and methods used (e.g., what does ‘we calibrate a general model’ mean?). I would also be good to know how relative deprivation and income rank are calculated as readers who are not experts on these topics might not know what these two concepts are reflecting.

We agree that it is important to be clear about the dataset and methods, especially about how income rank and relative deprivation are calculated. However, we were unable to do this fully within the constraints of a 150-word abstract. We have, however, now specified in the abstract that we use the Gallup dataset, made some other changes, and done our best to make sure that the critical concepts are introduced clearly and as early as possible in the revised manuscript.

By “we calibrate a general model,” we mean that we estimate the parameters of a general model (Equation 3, p. 5) using maximum likelihood estimation, which allows us to compare the effects of income rank and relative income deprivation on subjective well-being (SWB) within a single, unified framework.

Income Rank Calculation: This is defined in Equation 1 (p. 3) and further detailed in Equation 2 and 3 (p. 5).

Rank is operationalized as the ordinal position of an individual's income within their national distribution, scaled between 0 and 1.

Relative Deprivation Calculation: We distinguish this concept from rank-based models both theoretically and mathematically, especially in the section beginning on p. 3 and visualised in Figure 1 (p. 4). We adopt and extend established formulations such as the Yitzhaki Index, and estimate alternative models by adjusting weighting parameters (γ , δ) in Equation 3 to reflect deprivation-based weighting (pp. 4-5).

Point 2. Introduction

Point 2.1. SWB stands for 'Subjective wellbeing' not self-reported wellbeing as stated in the first paragraph of the introduction.

We thank the reviewer for pointing this out and have corrected it.

Point 2.2. The introduction is overall good with good detail about the studies.

We thank the reviewer for the positive comments.

Point 3. Results

Point 3.1. Study 1 should include sadness as another indicator of negative affect.

We thank the reviewer for the suggestion to include sadness as an additional indicator of negative affect. We have now incorporated sadness into the negative affect index in Study 1 and updated the manuscript text, relevant analyses, and figure descriptions accordingly.

The results are shown in Table 2, and, consistent with other results find that only income rank (and not absolute income) predicts negative affect.

Point 3.2. Study 1. This result should be explained further “Although some panels indicate a negative association between absolute income and affect levels, this association is small and non-significant in earlier survey waves” why might this be the case?

The negative association is no longer present (in the updated results using the latest-available survey). While the association appears in Round 12 (2017–2018) and a similar effect arises in Round 13 (2018–2019), as presented in the earlier version of our manuscript, it is not statistically significant in earlier or later rounds and so we do not consider the result robust. In line with the journal’s policy on null effects, and to avoid overinterpreting unstable findings, we refrain from drawing conclusions based on this association.

Point 3.3. Study 1. It looks like the mediation results are based on adding positive and negative affect as additional covariates. Although this method can tell whether a variable can be a potential mediator, this cannot be called mediation analyses and some test like Structural Equation Modelling should be conducted.

Thank you for this helpful comment. We agree that the analyses we included—adding affective and eudaimonic well-being variables as covariates—do not constitute a formal mediation analysis. As noted, methods such as structural equation modelling or causal mediation approaches would be required to make such claims.

Because mediation was not the focus of the study, and our intent was only to explore the relationships between variables descriptively, we have removed all references to mediation from the manuscript and supplementary materials. This change helps maintain the

clarity of our contribution, which is focused on the association between income, income rank, and SWB.

Point 3.4. Study 2. Was the test in Study 2 also run with pooling countries as in Study 1?

Thank you for the question. No, unlike Study 1, which pools data across countries to estimate overall effects, Study 2 only estimated the generalised model (Equation 3) separately for each country. This was necessary because the model includes country-specific income distributions and parameters (e.g., δ , γ) that capture potentially heterogeneous social comparison processes. Pooling would therefore obscure these differences. We have clarified this distinction more explicitly in the manuscript to avoid confusion.

Point 3.5. Study 3 and 4 are good and clearly explained although some concerns about waves still remain.

We thank the reviewer for the positive assessment of Studies 3 and 4. We understand and appreciate the concern regarding our reliance on a single survey round (Round 13, 2017–2018, in the original manuscript) for these analyses. This decision was driven by data availability: many of the moderators we examine—such as economic competitiveness, openness indicators, social capital measures, institutional trust, and economic preferences—are only available for a limited number of years. These variables are not consistently collected across all survey rounds, making it infeasible to conduct moderation analyses that span the full dataset. Our updated analysis (now focusing on Round 18, 2023–2024) includes both the SWB outcomes and access to the widest and most recent set of relevant moderators.

We also wish to highlight the issue of temporal mismatch. Using earlier rounds of SWB data would require pairing them with moderator variables measured in later years—for example, explaining 2014 SWB using

2017 competitiveness data. This would raise concerns about reverse causality: earlier well-being levels could plausibly influence later perceptions of institutional quality or economic conditions, rather than the other way around. This would undermine the interpretation of any moderation effect. By focusing on Round 18, we ensure that all moderators are measured prior to the well-being outcomes, which reduces ambiguity about the direction of influence.

We have clarified both the data availability constraints and reverse causality rationale more explicitly in the Methods section.

Point 4. Discussion

Point 4.1. Summary of results and limitations are well described. More on psychological mechanisms is needed.

Thank you for this suggestion. Although space limitations prevent a lengthy treatment, we have added (in the Discussion section at the end of the manuscript) some additional material about the relevant underlying psychological mechanisms.

Point 5. Methods

The article uses data from the Gallup World Poll (GWP) and there are a few points that need further clarification.

Point 5.1. It is not clear why the data used is 2009-2019 if the GWP goes from 2005 to 2024 already.

We have now updated the result throughout using the most recent available survey years. This has the advantage that a larger number of countries can be used.

We appreciate the reviewer's question. In the original

version of the paper, we analysed data from six rounds of the GWP covering a decade from 2009 to 2019: specifically, Rounds 4 (2009–2010), 5 (2010–2011), 8 (2013), 9 (2014–2015), 12 (2017–2018), and 13 (2018–2019). These rounds were selected to create three evenly spaced pairs across the decade, allowing us to assess both short-term (consecutive rounds) and medium-term (wider-interval) consistency of income rank effects on well-being.

To ensure cross-round comparability, we included only countries with complete data across all six GWP rounds, thereby forming a balanced panel. This approach helps reduce the risk that findings are driven by changing country coverage or sample composition across years.

In the updated manuscript, we extended the analysis to cover the most recent decade available, now spanning from 2013 to 2024: Round 8 (2013), 9 (2014–2015), 12 (2017–2018), 13 (2018–2019), 17 (2022–2023), and 18 (2023–2024). As before, we restricted the sample to countries with complete data across all included rounds to maintain longitudinal consistency. Our main results and conclusions remain both consistent and robust when tested against this updated dataset.

Point 5.2. A minor point is that the years might be better labelled as ‘survey years’ instead of ‘waves’ because ‘waves’ is usually used for longitudinal data and the GWP is cross-sectional.

We thank the reviewer for this helpful clarification. We agree that “survey years” is a more accurate term given that the GWP consists of repeated cross-sectional surveys rather than longitudinal panels. We have updated the manuscript accordingly by replacing “waves” with “survey years” throughout. When referring to Gallup’s internal survey cycle numbering, we now use the term “rounds” (e.g., “Round 13”), as a way to avoid the potentially misleading term “waves,” which Gallup uses internally.

Point 5.3. The methods section says “Social Status, Relative Deprivation, and Social Capital: The Ubiquity, Purity and Robustness of Income Rank Effects on Well-being in 90 Nations.” It is not clear how the authors came to this information because the GWP contains data from 165 countries and territories and more than 2 million respondents now.

Thank you for your comment. We appreciate the opportunity to clarify this point.

As noted in our response to point 5.1, the original version of the paper included data from 90 countries. These were selected based on the availability of data across six specific GWP survey rounds: rounds 4 (2009–2010), 5 (2010–2011), 8 (2013), 9 (2014–2015), 12 (2017–2018), and 13 (2018–2019). We chose these rounds because they form three evenly spaced pairs over a ten-year span, allowing us to examine the consistency of income rank effects on well-being both in the short term (between consecutive rounds) and in the medium term (across wider intervals).

To ensure comparability over time, we restricted the sample to countries with complete data in all six rounds, creating a balanced panel. This approach helps rule out the possibility that our findings are driven by shifts in country coverage across survey years.

In the updated analysis, we extended the analysis to cover the most recent decade available, now spanning from 2013 to 2024, beginning from round 8 (2013) and continuing through rounds 9, 12, 13, 17 (2022–2023), and 18 (2023–2024). Using the same criteria, we included only countries with complete data across these six rounds, resulting in a larger sample of 109 countries.

We hope this clarifies the rationale for the number of countries included in the analysis and the logic behind the panel construction.

Point 5.4. The methods section states “Each wave was analyzed independently and compared with the immediately preceding wave for robustness checks.” It is not clear why all survey years were not analysed together, and country and year fixed effects used in the regressions. If there is a clear rationale behind the choice of this method, this should be clearly explained.

We have now clarified our methodology and the rationale underlying it. Specifically, our aim is to exploit overall income differences between countries and hence distinguish between absolute and relative income more clearly than previous (fixed effects, in the case of multi-country) analyses have been able to. The use of fixed effects effectively re-introduces very high collinearity between absolute and relative income measures, thereby limiting interpretability. To mitigate potential confounding while preserving between-country variation, we have controlled for country-level differences far as possible.

More specifically: In Study 1, our primary goal is to directly compare the predictive power of absolute income versus relative income rank for SWB. Including country fixed effects would, by construction, remove all between-country variation in income – effectively transforming absolute income into a relative measure (i.e., income relative to others within the same country). In other words, absolute income would be turned into a within-country relative measure, much like income rank, and the model would no longer capture the independent effect of absolute income.

By analysing each survey round independently and avoiding country fixed effects, we retain both within- and between-country variation in income. This enables a conceptually valid comparison between absolute and relative income effects. In addition, comparing results across survey cycles allows us to assess the robustness of findings over time, while maintaining a clear interpretation of the income measures. We have revised

the text to explain this point more clearly.

Point 5.6. Why the moderation analyses were conducted only with 2018-2019 survey years should also be explained. Is it because the relevant variables are not available the previous years? In the methods section it says that this choice was made to ‘minimize the risk of potential reverse causality’, why is this the case?

Thank you for raising this point. As we outline in our response to point 3.5, the moderation analyses in Studies 3 and 4 were conducted using the 2018–2019 GWP survey data (Round 13) for two main reasons: data availability and concerns about reverse causality. We elaborate on each of these below.

Data availability: Many of the moderators we examine—such as economic competitiveness, openness indicators, social capital measures, institutional trust, and economic preferences—are only available for a limited number of years. These variables are not consistently collected across all survey rounds, making it infeasible to conduct moderation analyses that span the full dataset. Our updated analysis (now focusing on Round 18, 2023–2024) includes both the SWB outcomes and access to the widest and most recent set of relevant moderators.

Reverse causality concern: Using earlier rounds of SWB data would require pairing them with moderator variables measured in later years—for example, explaining 2014 SWB using 2017 competitiveness data. This raises concerns about reverse causality: earlier well-being levels could plausibly influence later perceptions of institutional quality or economic conditions, rather than the other way around. This would undermine the interpretation of any moderation effect. By focusing on Round 18, we ensure that all moderators are measured prior to the well-being outcomes, which reduces ambiguity about the direction of influence.

Point 5.7. Study 1 replicated prior work with an alternative method. The authors challenged the method widely used in prior work by saying that this method introduced multicollinearity when also including GDP per capita as a control and country fixed effects. For example, the authors say that “When studying multiple countries, previous research has typically employed country fixed effects to control for country-level differences and exploit within-country variations. However, this method introduces multicollinearity due to the high correlation among income-related variables.” If the issue is the within country-variation, the analyses should be conducted by each country not by each survey year or wave (I see an attempt of this in Study 2). This is because when conducting the analyses by survey year or wave, all the countries are still pooled together which is what the authors challenge in the method used in prior work. The findings presented in this paper should be replicated with this widely used method with and without GDP as a control.

We thank the reviewer for this observation. We appreciate the opportunity to clarify our approach and the focus of our critique regarding prior methods.

Our concern is not with pooling cross-country data or with analysing within-country variation per se, but specifically with the use of country fixed effects or GDP per capita controls when the goal is to estimate the effect of absolute income. These approaches, each in their own way, undermine the interpretability of absolute income and compromise the comparison with relative income.

Country fixed effects remove all between-country variation in income. As a result, the absolute income variable becomes centered within each country, meaning its coefficient reflects income relative to the national average rather than true individual-level material standards, as measured by income, across countries. In this context, absolute income effectively becomes a within-country relative measure, conceptually similar to income rank. This conflation makes it impossible to meaningfully compare the independent

effects of absolute and relative income, and introduces multicollinearity due to the high correlation among income-related variables.

Controlling for GDP per capita has a similar effect: even without fixed effects, it partials out cross-country income differences, reducing absolute income to a residualised measure of relative standing across countries. This again compromises the core distinction we aim to test.

Conducting separate analyses for each country, as suggested, would yield a similar challenge: absolute income and income rank would remain correlated within-country, making it difficult to isolate their distinct effects. That approach is better suited for examining purely relative income effects, which is precisely what we do in Study 2, where absolute income is no longer part of the analysis and the focus is on testing models of social comparison.

In Study 1, we instead analyse data separately by survey year, without country fixed effects, and include a rich set of country-level covariates. This approach allows us to preserve between-country variation in absolute income and within-country variation in income rank, enabling a conceptually valid and statistically identifiable comparison between the two. We have revised the text (see Methods) to clarify this rationale and distinguish it more explicitly from prior approaches.

Point 5.8. It would be good to know how the mediation analyses were conducted. This is important because the dependent variable and the mediators are all self-reported and reverse causality, multicollinearity, and endogeneity might be present.

We thank the reviewer for this comment. The mediation analyses in the original manuscript were intended as a brief, exploratory extension to examine whether

affective well-being or eudaimonic purpose might statistically account for some of the association between income rank and life evaluation. We did not interpret these results as causal and avoided making any such claims in the manuscript.

However, we agree that introducing mediation models – even when cautiously framed – may shift attention toward questions of mechanism that go beyond the core focus of the paper. Since the main goal of the manuscript is to test and validate the robustness and generality of income rank effects across contexts, we have decided to remove the mediation analyses in the revised version. We appreciate the reviewer's comment, which helped us refine the manuscript accordingly.

Point 5.9. In section 4.3 about Study 2, income rank and relative deprivation need to be defined (e.g., what do these mean). We know how these were calculated from the equation details, but we don't know what they mean in plain words.

We thank the reviewer for this observation. While the formal definitions of income rank and relative deprivation are provided in the equations and technical exposition, we agree that a plain-language explanation would help clarify the distinction for readers who are less familiar with these constructs.

We have now added brief, intuitive definitions in Section 4.3 of the revised manuscript. Specifically, we explain that:

Income rank refers to a person's position in the national income distribution – that is, how high or low their income stands compared to others in their country, scaled between 0 (lowest) and 1 (highest).

Relative deprivation reflects the extent to which a person falls short of those who are richer than them. It captures how much better off others are, on average,

and reflects upward comparisons.

Relative deprivation takes into account not just the number of higher earners, but the size of the income gaps.

We hope these additions improve the accessibility of the paper for a broader readership.

Point 5.10. Study 3 is well explained but the use of only ‘wave 13’ should be better justified and robustness checks with all the survey years together conducted.

Thank you for raising this point again. As noted previously (Point 5.6), we focused on the 2018–2019 survey year (Round 13) in Study 3 because, at the time of writing the original manuscript, it included the most comprehensive set of moderator variables. Many key moderators—such as social capital indicators, economic competitiveness and openness measures, and institutional trust—were only available in or around 2017–2018. This lack of consistent coverage makes it infeasible to replicate the moderation analyses across all survey years. Moreover, attempting to pair early survey years with moderator data from later periods would introduce serious concerns about reverse causality, compromising the interpretation of moderation effects.

To provide an updated validation of our findings, we re-ran the analysis using Round 18 (2023–2024), the most recent available data. The manuscript now focuses on this round, which includes both SWB outcomes and the most comprehensive and current set of relevant moderators. Our main results and conclusions remain consistent and robust when tested against this updated dataset.

In particular, the moderation analyses, at both the

individual and country levels, continue to show that social capital variables reduce the effect of income rank on well-being. We observe negative interaction effects with measures of social support, civic engagement (e.g., community involvement and willingness to volunteer or help others), and attitudes toward migrants. These findings replicate well, confirming that the moderation effects observed in Round 13 are not idiosyncratic to a single year, but are also evident in the most recent available data.

One exception is the interaction with market openness, which was significant in the original analysis but is no longer significant in the updated results. While market openness was not a primary focus of our original analysis, we report the updated result here for completeness. In the original manuscript, we used the Legatum Global Index of Economic Openness (2019). In the revised analysis, we draw on the 2023 Legatum Prosperity Index, which covers the same thematic pillars of economic openness but reflects a revised methodology. These changes, as well as broader shifts in economic conditions following the COVID-19 pandemic, likely account for the difference in findings and limit the comparability of openness indicators across survey years.

We have clarified our approach in the Methods section.

Point 5.11. More detail is needed regarding each moderator used in Study 3 and Study 4.

We thank the reviewer for this suggestion. In response, we have expanded the Supplementary Material to provide more detailed descriptions of all moderators used in Studies 3 and 4. For each moderator, we now include the original data source (e.g., Gallup World Poll, World Values Survey, Global Preferences Survey), the year of data collection, the exact question wording or a concise description of item content, and any

standardization or transformations applied during analysis.

Point 5.12. Ubiquity, purity, and ecological fallacy should be defined in the first part of the methods section.

We thank the reviewer for this suggestion. In the revised manuscript, we have added brief definitions:

Ubiquity refers to the consistency of an effect across different countries and time points, in our case, whether income rank predicts SWB universally.

Purity refers to the unique contribution of income rank to well-being, after accounting for other income-related variables such as absolute income or relative deprivation.

Ecological fallacy refers to the mistaken inference that relationships observed at the group or country level necessarily hold at the individual level. We explicitly avoid this fallacy by analysing country-level and individual-level moderators separately and interpreting effects only within their appropriate level of analysis.

These definitions have been added near the beginning of the Methods section, where we introduce our analytic strategy.

Reviewer #1 (Remarks on code availability):

I didn't revise the code because I had to request access to do so.

This has been fixed.

Reviewer #2 (Remarks to the Author):

The paper contributes to our understanding of the relationship between

income and wellbeing, which is an actively debated topic in wellbeing studies. The authors first conclude that it is relative income within a given country, rather than the absolute level of income, that matters for wellbeing. They then explore various approaches to measuring relative income, finding that a simple rank-based measure best fits the data. Finally, the authors demonstrate how the relationship between income rank and wellbeing varies across countries and among individuals with different characteristics.

It is a important contribution that the simple measure of rank performs as well, at least globally, as more complex measures accounting for asymmetries (Study 2). Your heterogeneity analyses (Studies 3 and 4) offer almost encyclopaedic detail — I find the results on civic engagement particularly interesting. However, I believe the main finding of the paper comes from Study 1, where you conclude that relative income rank within a country matters more for wellbeing than absolute income levels. I would want to better understand how one should interpret this result in the context of your modelling approach.

We thank the reviewer for the positive comments and detail below how we have addressed the points regarding interpretation.

1. The level-of-income vs. relative-income result is derived from a pooled cross-country regression without country fixed effects. The model controls for household income and income rank at the same time. This model specification is unusual. A standard cross-country wellbeing model would include household income and country fixed effects. The coefficient on household income from a standard model is then interpreted as the relations between wellbeing and higher or lower income in a given country, which is effectively relative income. The authors remove country fixed effect and include income both as a level (which is not significant) and as a percentile of income distribution (which is significant). The latter, effectively, captures similar variation to the coefficient on income from a model with county fixed effect. As a results, though the model formulation is unusual, I do not find the result surprising.

Thank you for your observation regarding our model specification. We would like to clarify the rationale behind our approach. While including country fixed effects is standard practice in cross-country well-being models to capture within-country variation, it was essential to exclude them for our specific research question: distinguishing between the effects of absolute income and relative income rank.

By excluding country fixed effects, we ensured that the coefficient on household income reflects changes in absolute income across the sample, while the coefficient on income rank captures the effect of relative status within each country. Including country fixed effects would shift the interpretation of household income to reflect income relative to the national average within each country, making it impossible to interpret it as an absolute measure. This would have limited our ability to differentiate between the impacts of absolute income and relative social status.

Previous studies that include country fixed effects typically find significant associations between income and well-being. However, these findings primarily reflect the effects of relative income within countries. In that light, it is perhaps not surprising that our models—without fixed effects—also show a strong association between income rank and well-being.

What our results clarify, however, is that while income rank remains a robust predictor, absolute income plays a comparatively weaker role when both are modelled simultaneously.

To address potential confounding from broader country-level factors that influence well-being, such as access to public services, social infrastructure, and overall quality of life, we include a set of macro-level controls. These include health expenditure per capita,

the Gini index, unemployment rate, and the percentage of the population living in rural areas. These variables help account for structural differences across countries that might otherwise obscure the distinct contributions of income to SWB. Associated limitations are acknowledged explicitly in the revised manuscript.

2. In Table 1, R-squared in Model 1 (with income in levels) is three times higher than in Model 3 (with relative income), suggesting that a combination of income level and wellbeing fits the data better than a combination of income rank and wellbeing. It would be helpful to see a model that includes both income variables without additional controls. I would expect the coefficient on log income in this model to be positive, because having higher income conditional on a given income rank implies living in a wealthier country. If it is not, it would be helpful if you could provide some intuition on why that is the case.

We thank the reviewer for this comment. In response, we have added a new set of models in Table S9 (Supplementary Material), where both log income and income rank are included as predictors without any additional controls (Column 3). We also present three additional models that sequentially introduce covariates, allowing readers to observe how the inclusion of individual- and country-level controls affects the relative contributions of each income measure. Column 4 adds individual-level controls, including age, gender, and a four-degree polynomial of age interacted with gender. Column 5 introduces country-level controls that have low correlation with log GDP per capita. Column 6 adds country-level controls that are highly correlated with GDP per capita (correlation > 0.7).

In line with the reviewer's intuition, the estimates in Column 3 (no controls) show that absolute income has a statistically significant effect, while income rank does not. However, it is important to note that in

models without controls, the coefficient on absolute income may partly reflect broader country-level differences that also influence SWB, such as access to public services, social infrastructure, and overall quality of life. Without adjusting for these contextual factors, the apparent predictive power of absolute income may be inflated, as it captures not only individual purchasing power but also national-level advantages that co-vary with income.

By introducing country-level controls in Columns 5 and 6, we aim to more precisely isolate the role of income itself, rather than broader country-level advantages correlated with it.

We have added a discussion of this analysis to the Robustness Analyses subsection of the Supplementary Material and incorporated a summary into the Results section of Study 1. We believe this addition clarifies the statistical relationship between absolute and relative income across different model specifications.

3. I assume that the coefficient on individual income in Model 5 is not significant, because the wellbeing benefits of leaving in a richer country are captured by controlling for health expenditures (correlation with GDP of 0.96) and absolute redistribution (correlation with GDP of 0.76). It is not clear to me how to select a set of country-level controls that allow controlling for heterogeneity without controlling away the effects of leaving in a wealthier country. It would be helpful to see the sensitivity of the results in Column 5 of Table 1 to including subsets of controls, e.g. individual level only, country-level with low correlation with GDP, etc.

We thank the reviewer for this comment. We would like to clarify that our individual-level absolute income variable (log of PPP-adjusted household income) is intended to capture a person's own material living standard, not the broader effects of living in a wealthier country.

In response to the reviewer's suggestion, we have added the requested sensitivity analyses in Table S9 of the Supplementary Material. Specifically, we observe that when macro-level variables are excluded, the coefficient on absolute income becomes statistically significant. However, this likely reflects the fact that, in the absence of such controls, absolute income partially captures national-level differences in infrastructure, services, and overall development-factors that go beyond an individual's purchasing power. In other words, without contextual controls, the absolute income coefficient conflates individual material resources with broader national living standards, making its interpretation ambiguous.

As shown in Table S9, once country-level controls are included, income rank consistently emerges as a stronger predictor of SWB than absolute income. This pattern supports our interpretation that relative social position plays a more prominent role in shaping well-being, once basic needs and broader contextual conditions are accounted for.

Although some of the country-level controls we include (e.g., health expenditure and redistribution measures) are correlated with GDP per capita, we retain them because they capture specific, policy-relevant dimensions of national context (such as public service provision and social safety nets) that may independently affect SWB. Using these targeted variables, rather than GDP per capita, allows us to retain meaningful between-country variation in absolute income without transforming it into a fully relative measure.

Importantly, after adjusting for both income rank and these macro-level contextual factors, absolute income no longer significantly predicts SWB. This highlights the stronger and more consistent role of social comparison once structural conditions are taken into

account.

We have clarified this point in the Robustness Analyses subsection of the Supplementary Material and in the Methods section of Study 1 (Section 4.2), and we appreciate the reviewer's suggestion, which prompted a valuable extension of our analysis.

4. The authors state that the magnitude of the coefficient on the income rank is high. One can interpret the magnitude of this coefficient as a wellbeing difference between the poorest and the richest household in a country. Then, the coefficient of 1.5 on a 0 to 10 scale in Column 5 of Table 1, though still large, does not look enormous. It would be helpful to see some intuition on how to compare relative magnitudes of the coefficients on income level and income rank, e.g. the expected effect of moving from the 25th to the 75th percentile according to the estimates from Columns 1 and 3.

We appreciate this helpful suggestion. To improve the interpretability of the coefficients in Table 1, we have added contextual comparisons for both absolute income (Column 1) and income rank (Column 3). As now described in the paper, absolute income is modeled as $\log(\text{income}/1000)$. In Column 1, the coefficient of 0.66 implies that moving from the 25th to the 75th percentile of income (approximately \$4,413 to \$26,188) is associated with a 1.18-point increase in life evaluation on a 1-10 scale.

In contrast, Column 3 shows a coefficient of 1.39 for income rank (ranging from 0 to 1), representing the estimated difference in well-being between individuals at the very bottom and top of the national income distribution. This rank-based effect is larger than the effect of a substantial change in absolute income.

While the magnitude may not appear extreme on its own, we note that it exceeds the effect sizes of several well-established socioeconomic variables in Column 4. For example, it is more than twice the size of the difference between having a college degree versus elementary education, and more than four times the difference between being single and separated. It is also roughly equivalent to twice the difference in life evaluation between being unemployed and being employed full-time. These comparisons help to place the rank coefficient in context and show that its effect, while not extreme in absolute terms, is substantively large relative to other common predictors of SWB. We have added these comparisons to the main Results section of Study 1 to improve interpretability for readers.

5. I don't believe that your findings corroborate the following statement in the conclusion: "The zero-sum property implies that, to the extent that a given domain of SWB is affected by income rank rather than absolute income, neither general increases in societal income nor redistributive policies will, in themselves, increase SWB." Firstly, given that your model effectively controls for GDP per capita, which can be interpreted as the level of societal income, I believe that a non-significant coefficient on individual income does not prove that general increases in societal income will not increase wellbeing. On redistributive policies, given that your analysis is cross-sectional, it is not informative about (potentially asymmetric) wellbeing benefits and losses from gaining or losing rank for people in different parts of the distribution. Since your research does not speak to that, I don't think that it can be used to support a statement about potential (non-)effectiveness of redistributive policies.

We thank the reviewer for this well-justified critique. We agree that the original statement in the conclusion was too strong, given the limitations of our data and design.

Specifically, because our models include controls for country-level variables that partially capture differences in infrastructure, services, and overall development, our findings do not directly speak to the well-being effects of increases in societal income.

Similarly, our cross-sectional design does not allow us to assess the potential asymmetric or dynamic effects of redistribution, such as gains or losses in rank across different parts of the distribution.

In light of this, we have revised the conclusion to more cautiously state that general income growth may have limited well-being effects when income rank is the dominant predictor, and that redistribution may still affect well-being through mechanisms not tested here (e.g., social cohesion, fairness, inequality perceptions).

We thank the reviewer for prompting us to refine the scope and interpretation of our findings.

Minor comments:

1. Wellbeing literature that uses comparator group typically defines it as a group with similar characteristics (e.g. age, gender, education, region). Could you please discuss why you believe national comparison to be the best choice in your case. Especially given that using subgroups of the population might also help you to alleviate concerns of within-country multicollinearity between income level and income rank.

We thank the reviewer for this comment. We agree that some studies of social comparison and SWB focus on reference groups defined by demographic or contextual similarity, such as age, gender, education, or local region.

In our study, we define income rank at the national level for both practical and conceptual reasons. First,

the GWP is designed to be nationally representative, but not necessarily representative within finer-grained subgroups. Constructing income rank within small subpopulations, such as by region, age, or education, would introduce considerable statistical noise and arbitrariness in group definitions. Rankings within small groups are less stable and potentially less meaningful, particularly in cross-national data where subgroup structures vary substantially. By contrast, national-level income rank offers a consistent and comparable reference frame across countries and years.

Second, while individuals may indeed make local comparisons in everyday life, national-level reference points are often more prominent in public discourse. National benchmarks, such as poverty lines, income deciles, and inequality indicators, are widely used in media, politics, and policy debates. These framings help make national income rank a psychologically and socially salient comparator, even if it is not the only one people use.

We also note that our modeling approach in Study 2 addresses this concern more directly. There, we implement flexible models of social comparison that allow the weight of others' incomes to vary based on their distance from an individual's own income. These models help capture whether individuals are more sensitive to close versus distant incomes, or whether they respond differently to those who earn more versus less. This framework can accommodate both national and more local comparison processes, and provides a complementary test of the underlying mechanisms.

We have added a short discussion of this rationale to the revised Methods section (Section 4.2) to clarify our choice of national income rank in light of our data structure and research design.

2. I am concerned about negative coefficient on the level of income in models for positive affects will all controls, both in the main result (Table 2) and in wave 12 (Table S5). Could you please provide some intuition on why this might be the case?

We thank the reviewer for highlighting this unexpected result. As noted, in fully specified models, which includes covariates at both the individual and national levels, the coefficient on log income becomes negative for positive affect in Round 12 (2017–2018) and a similar effect arises in Round 13 (2018–2019), as presented in the earlier version of our manuscript. This pattern is consistent across these two rounds but is not observed in earlier survey years or in more recent rounds (e.g., Rounds 17 and 18), as shown in the updated version of the manuscript. In line with the journal's policy on null effects, and to avoid overinterpreting unstable findings, we refrain from drawing conclusions based on this association.

3. It would be helpful to see if your main results in Table 1 hold for the equivalence-scale adjusted household income, since it is more accurately represents the level of income individual enjoys.

We agree that equivalence-scale adjusted income provides a more precise measure of the economic resources available to individuals within households. In response to this comment, we conducted a robustness check using a per capita income variable provided by Gallup. This measure is calculated by dividing total annual household income by the number of household members.

In Table S9, we report our main sensitivity analysis using the standard household income variable (PPP-adjusted, but not adjusted for household size). In Table S10, we replicate the same model specifications using per capita household income in place of the raw household income.

When we constructed income rank based on equivalised income, we found that this adjusted rank no longer significantly predicted SWB. This contrast suggests that individuals derive their sense of social status from their rank in the distribution of unadjusted income. That is, based on how their raw income compares to others rather than from a rank that reflects household-adjusted purchasing power.

While equivalized income more accurately captures material well-being, social status is shaped by perceived relative standing, which appears to be anchored in unadjusted income.

This likely reflects the fact that raw income is a more visible and culturally salient metric: it is how income is typically discussed, compared, and institutionally encoded (through tax brackets, salary negotiations, job titles, and public discourse). In contrast, equivalized income requires knowledge of household composition, which is neither intuitive nor socially salient.

These findings support the view that the effects of income rank on SWB are driven by social comparison and status-based mechanisms, rather than material conditions per se. Unadjusted income rank appears to better capture how individuals perceive their place in the social hierarchy, and it is this perception, rather than objective economic capacity, that predicts SWB. We have added a discussion of this robustness check and its implications to the Robustness Analyses subsection of the Supplementary Material.

Thank you for your work.

Reviewer #2 (Remarks on code availability):

I could not access the codes because the files linked above, https://osf.io/bj9cz/?view_only=8e0e7ff929e34cdab499695ef5843796,

require submitting a request for permission to gain access.

This is now fixed.

Reviewer #3 (Remarks to the Author):

Comments on Social Status, Relative Deprivation, and Social Capital: The Ubiquity, Purity and Robustness of Income Rank Effects on Well-being in 90 Nations Submitted to Nature Communications NCOMMS-24-42657-T

This paper studies the relationships between individuals' absolute income and relative income and their subjective wellbeing. We know that wellbeing increases with income (and that the effect is in part causal) but it is an ongoing topic of research as to how much is due to absolute vs relative effects. The absolute component is the degree to which an individual has more resources at their disposal. The relative component reflects the fact that a higher income might imply a greater position within a social hierarchy. If relative effects are dominant, this has some troubling implications for society, as it implies that economic growth (rising absolute incomes everywhere) may be limited in their ability to produce greater wellbeing.

The research here is an extremely comprehensive empirical exercise looking at the associations between income and income position (and a couple of related/derived measures) with wellbeing across 90 countries. The main innovation comes from the breadth of data used, and the recognition that country fixed-effects (normally a positive feature of these models) is actually a liability within this context. The reason is that an individual's income and their income rank (which in turn capture the absolute and relative components) are very highly correlated (the authors quote a correlation coefficient of 0.98). This means that it is immensely difficult to disentangle the effects of each, as there aren't many examples in the data where an individual's absolute income is much higher or lower than what would be implied by their rank.

The pooling of many country-level datasets offers a partial solution as

ranks get defined within countries while income gets a singular definition across countries. So by comparing people with \$X in poor countries and in rich countries (where the absolute value is constant, but the relative position changes) the authors can resolve this problem. The note that this pooling introduces a lot of heterogeneity, which they resolve with an extensive set of controls. The authors find considerable support for income ranks as a dominant driver of wellbeing, and distinguish this from a related concept of relative deprivation. They also find that income ranks are less important in countries with high levels of civic engagement, and uncover suggestive evidence that countries with more neoliberal politics might place more emphasis on relative effects.

This is a very strong paper. A high level of technical and conceptual expertise is shown throughout. The writing is careful, clear, and well-structured. I believe the paper is suitable for Nature Communications but have made some comments and suggestions below. All should be taken seriously, but I don't feel any are particularly critical.

We thank the reviewer for their positive evaluation and their helpful comments. We address the comments below.

- The analysis hinges very strongly on how well the country-level controls remove heterogeneity that could be correlated with income and may potentially explain wellbeing. While we can't tell for sure whether or not the controls are sufficient, if the results are heavily dependent upon a particular control, this may imply that that underlying factor isn't properly adjusted for. The authors could examine this further. It is a bit difficult to see what is happening as they just report "individual controls – yes/no" (and the same for countries). And sometimes the results bounce around and sometimes they don't.

We agree that it is important to assess how robust our findings are to different sets of covariates, particularly at the country level. In response to this comment, which was also raised by Reviewer 2, we have expanded our analysis to include a stepwise specification approach. We now present models that

sequentially add controls: starting with individual-level covariates (e.g., age, gender, education, marital status), followed by blocks of country-level controls that have low correlation with GDP per capita, and finally adding those with high correlation (e.g., absolute redistribution, urbanization levels).

This breakdown is now presented in a revised supplementary table (Table S9), which shows how the coefficients on income level and income rank change as additional controls are introduced.

Unlike the original version of the table, where control blocks were grouped using "yes/no" labels, the revised table displays the individual coefficients for each control variable. The only exception is the set of age-related controls: models in Columns 4 to 6 include a four-degree polynomial of age interacted with gender. Due to the large number of resulting terms, we opted not to report these coefficients in the table itself, but we clearly note their inclusion in both the Methods section and the table note.

As shown in Table S9, once country-level controls are included, income rank consistently emerges as a stronger predictor of SWB than absolute income. This pattern supports our interpretation that relative social position plays a more prominent role in shaping well-being, once basic needs and broader contextual conditions are accounted for.

We believe this revised table provides a clearer and more detailed view of how specific controls influence the relationship between income and well-being, and we thank the reviewer for this helpful suggestion.

- The authors don't use the concepts of exogeneity/endogeneity at all in the paper. I appreciate that this is a general interest journal, and the readership may not be versed in econometric terminology, but these ideas are too useful to just gloss over. Since the pdf is 75 pages long it would be

easy to briefly explain these concepts and make it clear to the reader that the causal claims are fairly muted.

We thank the reviewer for this important point. While we aimed to maintain accessibility for a general audience, we fully agree that concepts such as exogeneity and endogeneity are critical for framing the limits of causal interpretation in observational data. In response, we have now added a short section in the Discussion where we explicitly discuss the potential for endogeneity in our models, including issues such as omitted variable bias, measurement error, and reverse causality. We clarify that, while our models include extensive individual- and country-level controls, they do not identify causal effects in the strict econometric sense.

To address this, we now describe our results as reflecting associations rather than effects unless otherwise specified, and we explicitly state that causal claims should be interpreted with caution. We also provide a brief explanation of endogeneity and exogeneity for readers who may be unfamiliar with these terms, and note that future work using panel or experimental designs would be needed to establish causal pathways.

These revisions appear in the updated Discussion.

- Following on from the above point, the authors should probably further mute their causal claims and tone down the causal language.

Thank you for the comment. We have revised the manuscript to mute the causal claims and tone down the causal language throughout the text.

- I have my doubts about using the indicator approach used to handle missing data. This is a simple method where a missing value for a covariate is imputed (usually with the mean, but it doesn't matter), and an

additional dummy variable is included in the regression to indicate that a value was imputed. This is only done for the country-level data, but it can be a source of bias in non-experimental studies (e.g., Groenwold et al., 2012). Does this improve statistical power? Maybe, because you get more data, but maybe not, because you have to estimate more covariates. Why not use a standard multiple imputation technique? Or just report results for the data you have?

We appreciate the reviewer's concern regarding our use of the missing-indicator method to handle missing observations in country-level covariates. For covariates with missing values, we set the missing entries to zero and included a corresponding dummy variable indicating that the value was originally missing. This approach allowed us to retain countries with incomplete data while accounting for the presence of missingness in the model. We adopted this strategy to preserve comparability across models and avoid further reducing the number of countries, which is already limited in some parts of the analysis. However, we fully acknowledge that this method has limitations and may introduce bias if the data are not missing at random.

To address this concern directly, we now report results from a complete-case analysis, restricting the sample to countries with no missing values in country-level covariates (as well as no missing data at the individual level). These results are presented in Supplementary Table S8, which replicates our main analysis from Table 1 (Study 1) using the reduced dataset.

The results are highly consistent with those in the main specification. The effect of income rank remains statistically significant and is slightly larger in magnitude, and the relative magnitudes of other key predictors remain stable. We now reference this robustness check in both the Methods section and

Results to provide transparency and reassurance regarding our approach.

- Out of interest, what happens to the estimates in Table 1 when the fixed-effects are included?

While we believe that including country fixed effects complicates the interpretation of the absolute income coefficient, since all between-country variation is absorbed and the effect is estimated solely from within-country differences, we agree that presenting these models is informative for comparison purposes.

In Supplementary Table S7, we therefore now replicate the main results from Table 1 (Study 1) with country fixed effects included. As expected, all country-level covariates are omitted due to multicollinearity. The coefficient on income rank remains consistent and statistically significant in the fully specified model (last column). However, the coefficient for absolute income becomes negative and non-significant. This likely reflects the fact that, in the presence of fixed effects, absolute income is effectively centered within countries. As a result, its coefficient no longer captures variation in individual-level absolute income across countries, but rather income differences relative to each country's mean. This redefinition makes it conceptually similar to income rank and introduces substantial multicollinearity between the two variables ($r \approx 0.97$), which inflates standard errors and limits interpretability.

We reference these results in the revised manuscript and Supplementary Material and thank the reviewer for prompting this analysis.

- It is tempting to think that relative effects will become larger in richer countries, which the authors demonstrate. But do absolute effects dominate in poorer countries? Surely a starving child cares much more

about food than they do about social status. This would be simple to carry out and could be reported in an appendix. If true, it would also serve as a rough intuition check for the rest of the paper.

Thank you for this suggestion. We have re-estimated the models from Table 1, splitting the sample at the median GDP per capita to distinguish between poorer and richer countries. The results are reported in Table S11 in the Supplementary Material. Columns 1-3 present results for countries below the median GDP per capita, while Columns 4-6 present results for those above the median.

As anticipated, the effect of absolute income is larger in the poorer-country subsample, where income is more directly linked to quality of life through its role in meeting basic needs. In contrast, the effect of income rank is stronger in the richer-country subsample, where income tends to affect well-being more through discretionary spending and relative social standing.

Notably, however, the income rank variable remains statistically significant in the full model (including controls for demographics and country characteristics) in both groups. In contrast, the absolute income effect, although larger in magnitude for poorer countries, becomes statistically non-significant once controls are included. This finding supports your intuition that income rank may play a relatively greater role in shaping life evaluation in wealthier contexts.

- Some of the tables are too big and contain too much information to be easily interpretable.

We thank the reviewer for this helpful comment. We acknowledge that some tables, particularly those in Study 1, are extensive and may be dense to read. This reflects our decision to display the full set of control variables, rather than summarising them with

binary indicators (e.g., "YES/NO"), in order to enhance transparency. We appreciate the reviewer's earlier suggestion on this point and have aimed to strike a balance between clarity and completeness—especially in our stepwise robustness checks, where understanding the role of specific covariates is important.

To improve interpretability in Studies 3 and 4, we have enhanced the visual presentation of results by presenting marginal effects using plots that now also include comparisons with reference categories (i.e., the first variable within each group of covariates), making within-category comparisons easier. The corresponding regression tables are included in the Supplementary Material. We now clarify in the manuscript that the figures are directly linked to these tables, which should help guide readers while still providing full access to the underlying estimates.

We hope this approach offers a clear and accessible presentation of the findings while maintaining transparency in model specification.

- The fact that income ranks seem less important in countries with more social capital, and for individuals with more social capital, seems like a meaningful and under-discussed result. It is easy to see how the concept of endogeneity could be used to explain the latter (e.g., people with positive social characteristics may be autonomously happy and insensitive to income in general) but not the former.

We thank the reviewer for this observation. We agree that the finding, that the association between income rank and SWB is weaker in contexts with higher social capital, is both theoretically meaningful and deserves greater emphasis. In the revised manuscript, we expand the Discussion to highlight this point more clearly.

We now distinguish between individual- and country-

level moderation effects. At the individual level, it is possible that people with higher trust or stronger social connections are less sensitive to income comparisons because of stable personality traits or emotional dispositions that also enhance SWB. This raises the possibility of endogeneity, in which the observed moderation effect may reflect underlying differences in individual temperament rather than the protective role of social capital per se.

However, the country-level moderation is less vulnerable to this concern. National levels of social capital, such as trust in others, civic engagement, and community support, reflect broader structural and cultural conditions that are unlikely to be influenced by individual SWB. This asymmetry strengthens the interpretation that higher social capital at the national level may buffer the psychological effects of income inequality. We now emphasize this point in the revised Discussion and we believe it highlights an important avenue for future work on the interaction between social context and the consequences of relative income.

- **The authors worry a little that their self-reported income data may not be reliable. If miserable people over-report their income (which seems more likely than the reverse) then this would attenuate the results. So the real correlations might be stronger than the ones reported here.**

We thank the reviewer for this observation. We agree that self-reported income may be affected by measurement error, which could attenuate associations with well-being, particularly for absolute income. We now acknowledge this as a limitation in the Discussion section. While our findings consistently show that income rank is a stronger predictor of well-being, we recognise that differential sensitivity to measurement error may partially contribute to this pattern. At the same time, we note that a central aim of the paper is to test a theoretical claim about the

psychological relevance of relative standing. This theoretical contribution remains valid, even if the magnitude of the absolute income effect may be underestimated due to measurement error.

- The depressing conclusion that income rank is zero sum could be expanded upon. The stakes of this research are quite high, and a better discussion of the unpleasant result and its social implications would help.

We have revised the conclusion to more cautiously state that general income growth may have limited well-being effects when income rank is the dominant predictor, and that redistribution may still affect well-being through mechanisms not tested here (e.g., social cohesion, fairness, inequality perceptions).

Reviewer #3 (Remarks on code availability):

I had a quick look at the code. It looks very professional, but I didn't scrutinize it for errors. Since the data is public it should be straightforward to reproduce all results.

Reviewer #4 (Remarks to the Author):

This is a very ambitious paper. I think the paper's most significant and interesting contribution is to tease apart the differing effects of income rank and relative deprivation--showing that income rank is by far a stronger measure (Study 2). The exploration of heterogeneous effects of income rank in Studies 3 and 4 is interesting as well and complements Study 2 nicely.

My largest concern is with study 1, which examines absolute vs relative income. The authors note the problem that "Within single countries, incomes and relative income ranks are highly correlated, making it difficult to separate their effects statistically." (p. 6). It seems the problem is "solved" by pooling countries, so effectively our variation is by comparing two people with the same income, but one in a poor country where they

have a higher rank) to someone in a rich country (where they have a lower rank).

But, the authors explain in the methods that "We measure income as annual household income in international dollars, calculated by the GWP using World Bank's PPP estimates, represented in 2011 US dollars." So, in my example above, two people with the same income (as measured in international dollars) have the same purchasing power, regardless of which country they are in. Thus, it's little surprise that they their income rank would have more influence than absolute income.

We thank the reviewer for raising this important conceptual point. While it is true that PPP-adjusted income equalises average purchasing power across countries, this adjustment does not eliminate meaningful variation in individual-level income, either within or across countries. PPP allows for cross-country comparisons in what a given income can buy, for example, comparing the purchasing power of a dollar in Kenya to that of a dollar in Canada, but it does not compress national income distributions or remove between-country differences in individual income levels.

In our dataset, PPP-adjusted income remains highly variable. The comparatively weaker effect of absolute income in our models is therefore not an artefact of PPP adjustment, but rather reflects the stronger explanatory power of relative standing (income rank) when both variables are included in the same specification. Additionally, the inclusion of country-level controls helps isolate the individual-level effect of income from broader contextual factors, such as infrastructure, public services, and inequality, that might otherwise inflate the apparent role of absolute income.

We appreciate the opportunity to further clarify the distinction between PPP-adjusted income and the social

status associated with income. While PPP ensures comparable purchasing power (meaning what a given income can buy), it does not account for the relative economic rank or the social and psychological implications tied to it. Two individuals with the same PPP-adjusted income may occupy very different positions in their national income distributions. For example, one individual with such income might hold a high relative rank and greater social standing in a lower-income country, whereas another might have a modest rank and lower perceived wealth in a higher-income country.

Conversely, individuals at the same income rank in different countries may have very different absolute incomes, reflecting variation in the underlying income distributions across countries. This underscores that income rank and PPP-adjusted income capture conceptually and empirically distinct aspects of economic experience.

Also related to Study 1, in the conclusion of the paper, the authors finish by stating that "...The zero-sum property implies that, to the extent that a given domain of SWB is affected by income rank rather than absolute income, neither general increases in societal income nor redistributive policies will, in themselves, increase SWB." (p. 19). But, note that the models from Study 1 have included measures such as log health expenditures per capita (which has a large positive effect on SWB, Table 1), and which has a .96 correlation with log GDP (Table S1). If a general increase in societal income is part of an increase in GDP or health expenditures (or whatever else these proxy for), then this likely does increase SWB.

We thank the reviewer for this comment, which was also raised by Reviewer 2. We agree that the original statement in the conclusion was too strong, given the limitations of our data and study design.

Specifically, because our models include controls for

country-level variables that partially capture differences in infrastructure, services, and overall development, our findings do not directly speak to the well-being effects of increases in societal income.

Similarly, our cross-sectional design does not allow us to assess the potential asymmetric or dynamic effects of redistribution, such as gains or losses in rank across different parts of the distribution.

In light of this, we have revised the conclusion to more cautiously state that general income growth may have limited well-being effects when income rank is the dominant predictor, and that redistribution may still affect well-being through mechanisms not tested here (e.g., social cohesion, fairness, inequality perceptions).

We appreciate the reviewer's comment, which prompted us to refine the scope and interpretation of our findings in the revised manuscript.

Other (more minor) comments:

* Social capital is in the title, but it plays a very minor role in the results (alongside with other potential mediators), and has no part in the background. There is no review of the extensive literature on how social capital is measured and how it may affect SWB. I think these are still fine analyses to have, but "social capital" certainly shouldn't be part of the title, as it's not what the paper is about.

We acknowledge this point. However we also note that Reviewer 3 highlighted this aspect as a meaningful and under-discussed result:: "The fact that income ranks seem less important in countries with more social capital, and for individuals with more social capital, seems like a meaningful and under-discussed result."

We have therefore added in a small amount of additional

discussion of the social capital results (see response to Reviewer 3) and have opted to retain the current title, which we believe reflects one of the key contributions of the paper.

* In study 1, the authors write: "These results show that (a) the rank of a person's income is a better predictor..." As far as prediction accuracy, the R-squared of both rank and absolute income are similar (and both are low). The authors mean something like a larger effect (although the units are different...)--regardless of how they want to say it, more care is needed here.

We thank the reviewer for pointing out this important distinction. Our original phrasing was imprecise and may have suggested that income rank explains significantly more variance in well-being outcomes, which is not supported by the R^2 values. What we intended to convey is that income rank shows a larger and more consistent association with well-being, and that when income rank and absolute income are both included as predictors, only income rank significantly predicts the outcomes.

To avoid confusion, we have revised the relevant sentence in Study 1 (p.10).

We believe the revised wording more accurately reflects the results and avoids overstatement.

* I like the moderation analyses in Studies 3 and 4, but the figures fall a bit short in conveying the key information. The evidence of an effect is the size of the gap between the two predicted marginal effects for each variable. This is hard to eyeball. I wonder if the authors could find a way to include this information as well?

Thank you for this suggestion. The revised plots for Studies 3 and 4 now display the moderation effects for different groups of indicators—such as demographics, confidence in institutions, and social preferences—on

the left side. The right panel shows the differences in effects relative to the reference category (the first category within each group). We believe this revised presentation facilitates clearer comparison of marginal effects within groups.

* In the supplement, the authors write "In our study design, we do not conduct a power analysis to determine our sample size, as we are not powering for null hypothesis significance testing." But really, as soon as you are reporting statistical significance tests, that's exactly what you are doing. The statistical significance test is based on the logic of null hypothesis testing. I don't necessarily feel that power analyses are needed, but this statement should be omitted at the very least.

We agree. We have removed this statement.

Reviewer #4 (Remarks on code availability):

I looked at some of the code (including the README), but the data are not available, so I did not run anything.

An updated version of the code incorporating the new robustness tests is now available.